# Three-Dimensional Structure of Wind Turbine Wakes as Measured by Scanning Lidar

Nicola Bodini[1,2], Dino Zardi[2], and Julie K. Lundquist[1,3]

[1]Department of Atmospheric and Oceanic Sciences, University of Colorado Boulder, Boulder, Colorado, USA
[2]Department of Civil, Environmental and Mechanical Engineering, University of Trento, Trento, Italy
[3]National Renewable Energy Laboratory, Golden, Colorado, USA

*Correspondence to:* Nicola Bodini (nicola.bodini@colorado.edu)

**Abstract.** The lower wind speeds and increased turbulence that are characteristic of turbine wakes have considerable consequences on large wind farms: turbines located downwind generate less power and experience increased turbulent loads. The structures of wakes and their downwind impacts are sensitive to wind speed and atmospheric variability. Wake characterization can provide important insights for turbine layout optimization in view of decreasing the cost of wind energy. The CWEX-13 field campaign, which took place between June and September 2013 in a wind farm in Iowa, was designed to explore the interaction of multiple wakes in a range of atmospheric stability conditions. Based on lidar wind measurements, we extend, present, and apply a quantitative algorithm to assess wake parameters such as the velocity deficits, the size of the wake boundaries, and the location of the wake centerlines. We focus on wakes from a row of four turbines at the leading edge of the wind farm to explore variations between wakes from the edge of the row (outer wakes) and those from turbines in the center of the row (inner wakes). Using multiple horizontal scans at different elevations, a three-dimensional structure of wakes from the row of turbines can be created. Wakes erode very quickly during unstable conditions, and can in fact be detected primarily in stable conditions in the conditions measured here. During stable conditions, important differences emerge between the wakes of inner turbines and the wakes of outer turbines. Further, the strong wind veer associated with stable conditions results in a stretching of the wake structures, and this stretching manifests differently for inner and outer wakes. These insights can be incorporated into low-order wake models for wind farm layout optimization or for wind power forecasting.

## 1 Introduction

A wind turbine wake is the volume downwind of a wind turbine, affected by the fact that the wind turbine removes momentum from the flow, thus reducing the downwind speed. Also, in the wake, the flow is more turbulent than in the inflow, because of the rotation of turbine blades and the presence of the wind turbine itself as an obstacle to the incoming wind flow (Landberg, 2015).

Wind turbine wakes impact the layout optimization and energy production of large wind farms (Brower, 2012). In fact, the reduced wind speed in the wake region has a direct effect on the power extracted by downwind turbines (Neustadter and Spera, 1985; Barthelmie et al., 2010; Nygaard, 2014). Moreover, the increased turbulence in wakes enhances turbulent loads

for downwind turbines, possibly inducing premature failure (Crespo et al., 1999; Sathe et al., 2013).

Therefore, wakes need to be studied and understood in order to maximize the efficiency of wind energy production. In particular, wake models are applied in several steps of the design and life-time management of wind farms, whose layout is studied in detail to maximize the amount of energy generated by the turbines (Elkinton et al., 2006; Samorani, 2013). Moreover, the overall wind resource assessment process needs to take into account the effect of wakes to have a reliable prediction of future power production (Brower, 2012; Clifton et al., 2016). Lastly, wind farm control techniques incorporate detailed studies of wake characteristics while periodically changing some features of the turbines (such as yaw angle and pitch angle) in order to maximize the overall power production from the whole wind farm (Fleming et al., 2014, 2016; Gebraad et al., 2016; Vollmer et al., 2016). Typically, all these processes are computationally intensive, and apply low-order models of turbine wakes (Elkinton et al., 2006; Chowdhury et al., 2012), such as the Jensen model (Jensen, 1983; Katic et al., 1986). In this way, several scenarios can be tested, but these lower-cost models oversimplify reality and may not be capable to fully represent wake characteristics in a detailed and realistic way (Barthelmie et al., 2006; Andersen et al., 2014).

Atmospheric stability has been shown to have a major impact on wind turbine wake evolution and wind farm performance in both observational (Magnusson and Smedman, 1994; Hansen et al., 2012; Wharton and Lundquist, 2012; Vanderwende and Lundquist, 2012; Barthelmie et al., 2013; Dörenkämper et al., 2015; Machefaux et al., 2015; Mirocha et al., 2015) and modeling studies (Churchfield et al., 2012; Aitken et al., 2014b; Mirocha et al., 2014; Machefaux et al., 2015; Abkar and Porté-Agel, 2015; Abkar et al., 2015): wakes in stable conditions persist for long distances downwind, while during unstable conditions the enhanced turbulent mixing erodes the wakes more quickly.

Wake characterization from field data can validate and improve the quality of numerical models. Data from field campaigns avoid possible limitations of wind tunnel simulations, such as down-scaled geometric dimensions and low Reynolds number (Iungo et al., 2013). Lidars and radars have been widely used recently to characterize wind turbine wakes. These instruments can measure wind characteristics above the heights of most traditional meteorological towers, and they can be deployed and moved rather easily, allowing measurements at several different locations. Many wake validation studies from remote sensing measurements focus on individual isolated turbines (Käsler et al., 2010; Bingöl et al., 2010; Trujillo et al., 2011; Hirth et al., 2012; Hirth and Schroeder, 2013; Aitken et al., 2014a; Aitken and Lundquist, 2014; Bastine et al., 2015; Kumer et al., 2015), with some studies that aim at reconstructing the 3-dimensional structure of wind turbine wakes (Iungo et al., 2013; Banta et al., 2015). The interactions between multiple wakes must be captured in studies of large wind farms, as done by Clive et al. (2011); Hirth et al. (2015a, b); Kumer et al. (2015); Wang and Barthelmie (2015); Aubrun et al. (2016); van Dooren et al. (2016).

In this paper, we analyze scanning lidar and profiling lidar measurements from the CWEX-13 field campaign in a large wind farm in Iowa, and we extend the individual wake detection algorithm proposed by Aitken et al. (2014a) to characterize multiple wakes. The three-dimensional structure of wakes from a row of four turbines is assessed, in terms of velocity deficit, width of the wakes, and wake centerlines. Section 2 describes the CWEX-13 field campaign and how we use measurements from the instruments deployed at the site. In Section 3 we present the wake characterization algorithm for multiple wakes, an expansion of the algorithm proposed by Aitken et al. (2014a). Section 4 highlights how wake characteristics (velocity deficit, wake width, and wake centerline) change in three-dimensional space, and for the first time we quantify the effect of ambient wind veer on

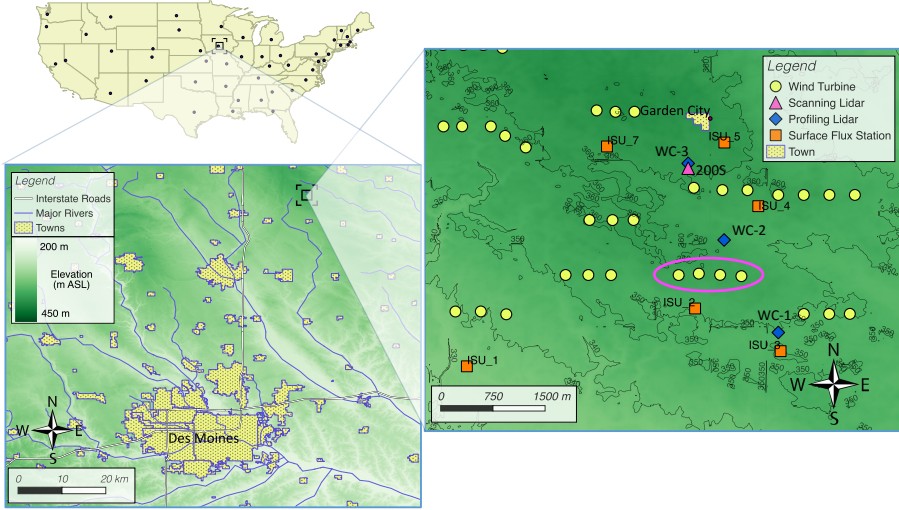

**Figure 1.** Schematic view of the part of the wind farm in central Iowa where the CWEX-13 field campaign took place. The row of four turbines whose wakes are detected by the scanning lidar is highlighted in a purple ellipse.

the vertical stretching of the structure of wakes. In Section 5 we compare the present results with those obtained in previous studies, and we suggest possible future work to improve wake simulations and models.

## 2 Data and Methods

This study analyzes the scanning lidar and profiling lidar measurements from the CWEX-13 observational campaign, summa-
rized in Takle et al. (2014) and Vanderwende et al. (2015).

### 2.1 CWEX-13 Observational dataset

CWEX-13 campaign (Lundquist et al., 2014) took place between late June and early September 2013 in a wind farm in central
Iowa, the same wind farm studied in previous CWEX campaigns; however, CWEX-13 focused on a part of the wind farm that
is different from what is discussed in Rajewski et al. (2013), Rhodes and Lundquist (2013), Mirocha et al. (2015), and Lee
and Lundquist (2017). The region exhibits strong diurnal cycles of atmospheric stability, and frequent nocturnal low-level jets
(Vanderwende et al., 2015). The area has a flat topography, with large fields of corn (height $1 - 2$ m) and soybeans (height
$0.3 - 0.8$ m). The region also has four small villages, some riparian regions, and a few trees and buildings.

Figure 1 shows a schematic diagram of the area of the wind farm of interest in CWEX-13. The yellow dots represent the
wind turbines, whose main technical specifications are reported in Table 1. For the purpose of this work, we focus on the
characterization, using scanning lidar data, of the wakes from the row of four turbines enclosed in the purple ellipse in Figure
1.

**Table 1.** Technical specifications of the studied wind turbines in CWEX-13 field campaign.

| | |
|---|---|
| Rotor diameter ($D$) | 80 m |
| Hub height | 80 m |
| Rated power | 1.5 MW |
| Cut-in wind speed | 3.5 m s$^{-1}$ |
| Rated power at | 11 m s$^{-1}$ |
| Cut-out wind speed | 20 m s$^{-1}$ |

### 2.1.1 Lidar measurements

Three WINDCUBE v1 vertical profiling lidars (blue diamonds in Figure 1) were deployed at the site during the field campaign, and they were located south of the studied row of four turbines, $8.5D$ north of the above-mentioned turbines, and $5.7D$ north of a second row of turbines. These instruments provided vertical profiles of wind speed and direction from 40 m to 220 m above the surface, with measurements collected every 20 m. At CWEX-13, southerly wind conditions dominated the campaign. So, we used data from the WC-1 profiling lidar to measure upwind conditions for the studied row of turbines, and calculate the ambient wind veer.

From 31 July to 6 September 2013, a LEOSPHERE WINDCUBE 200S scanning lidar was deployed with the northernmost WINDCUBE v1 profiling lidar (WC-3 in Figure 1). Vanderwende et al. (2015) demonstrated good agreement between the co-located scanning and WC-3 profiling lidars measurements at the altitudes where measurements overlapped. Scanning lidars can operate sweeping the azimuth angle with a constant elevation angle, the so-called *plan-position-indicator* (PPI) mode (*velocity azimuth display* - VAD - mode when a full conical scan is conducted), or sweeping the elevation angle while holding the azimuth angle fixed, in the so-called *range-height-indicator* (RHI), mode (Sathe and Mann, 2013). In CWEX-13, the scanning lidar used a combination of PPI, VAD and RHI scans, with a 30-minute cycle (each PPI scan lasted approximately 100 seconds, spanning an azimuth range of $50°$ with a speed of $0.5°\text{s}^{-1}$, while a RHI had a duration of about 30 seconds, see Table 2). Measurements were collected with slant range gates of 50 m at ranges up to 5000 m from the instrument, with an angular resolution of $0.5°$. Line-of-sight (radial) velocity was measured with an accuracy better than 0.5 m s$^{-1}$.

Given the dominant southerly wind conditions, the WINDCUBE 200S scanning lidar can use horizontal (PPI) scans, to observe wakes propagating from the row of four turbines of interest. The horizontal scans were performed at six different elevation angles, giving a range of different vertical positions depending on the distance from the lidar. Approximately ten minutes were required to collect the series of six elevation tilts; elevation angles varied from $1.5°$ to $2.8°$, which allow measurements at a variety of vertical positions between the bottom and top of the rotor disk of the turbines, as shown in Figure 2.

We select for a detailed analysis two days (23 and 26 August 2013) displaying wind conditions representative of the typical southerly wind pattern for the site. The first case, 23 August, had predominant southeasterly wind conditions, with relatively low wind speed, which never exceeded 10 m s$^{-1}$ at 220 m AGL. During 23 August 2013, 438 PPI scans were performed, 73

**Table 2.** Description of the 30-min cycle of scanning lidar scans in CWEX-13 field campaign. The characteristic fixed angle refers to the elevation angle for PPI and VAD scans, the azimuth angle for RHI scans.

| number of scans | type of scan | characteristic fixed angle | duration of each scan | cumulative time |
|---|---|---|---|---|
| 2 | VAD | 75°, 60° | 132 s | 0:00 - 4:24 |
| 6 | PPI | 2.8°, 2.5°, 2.2°, 2.1°, 1.8°, 1.5° | 104 s | 4:24 - 14:48 |
| 3 | RHI | 160°, 170°, 180° | 32 s | 14:48 - 16:24 |
| 6 | PPI | 2.8°, 2.5°, 2.2°, 2.1°, 1.8°, 1.5° | 104 s | 16:24 - 26:48 |
| 6 | RHI | 160°, 170°, 180°, 180°, 170°, 160° | 32 s | 26:48 - 30:00 |

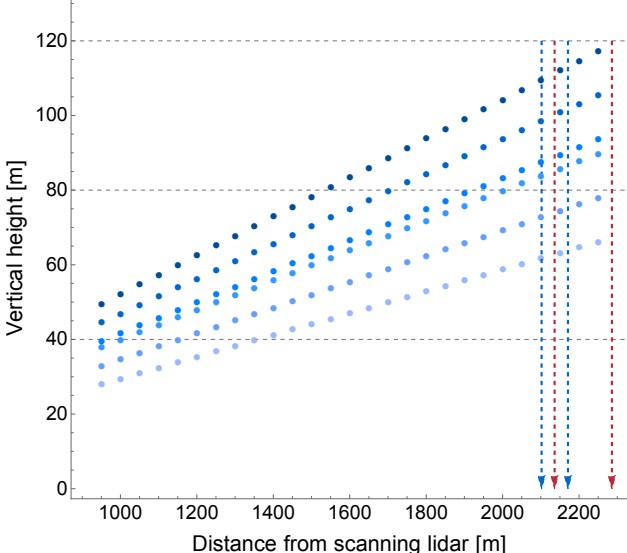

**Figure 2.** Schematic representation of the position where measurements from PPI scans are available, at six different elevation angles, as a function of the distance from the scanning lidar. The horizontal dashed lines show the vertical limits of the rotor disk of the turbines and hub height. The position of the four turbines is represented by the vertical dashed lines, with red lines for outer turbines and blue lines for inner turbines. From the westernmost to the easternmost turbine, the distances from the scanning lidar are 2136 m, 2102 m, 2171 m, and 2286m. The change in elevation between the turbine location and the lidar location (7 m) is taken into account.

for each of the six elevation angles. On the other hand, 26 August showed southwesterly wind, which is the most common situation for the site, with greater wind speed (up to 20 m s$^{-1}$ at 220 m AGL). During 26 August 2013, 576 PPI scans were performed, 96 for each elevation angle. By comparing the results from these two different days, the effect of wind direction on some wake characteristics can be assessed. Figure 3 shows examples of maps of line-of-sight velocity measured by the

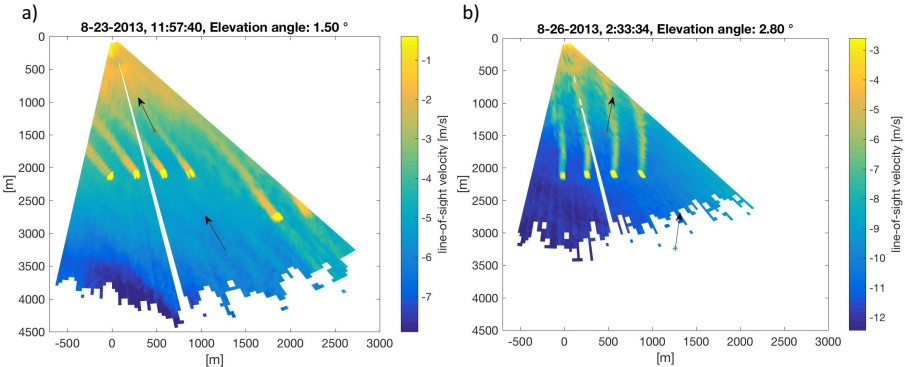

**Figure 3.** Color maps of line-of-sight velocity measured by the scanning lidar during two PPI scans performed at 11:57 UTC (6:57 LDT) on 23 August 2013 (panel a) and at 02:33 UTC (21:33 LDT) on 26 August 2013 (panel b). The scanning lidar is located in the origin of the coordinate system. The two arrows show wind direction as measured by the profiling lidars WC-1 and WC-2 at 80 m AGL.

scanning lidar during two PPI scans performed at night on the selected days. The wind turbine wakes can clearly be detected in terms of reduced wind speed downwind of the four wind turbines.

### 2.1.2 Surface flux measurements for quantifying atmospheric stability

Several surface flux stations (provided by Iowa State University) were deployed at the CWEX-13 site (orange squares in Figure 1). We used measurements from the surface flux station ISU_3 to assess atmospheric stability conditions, with the calculation of Obukhov length $L$, defined as:

$$L = -\frac{\overline{\theta_v} \cdot u_*^3}{k \cdot g \cdot \overline{w'\theta'_v}} \tag{1}$$

where $\theta_v$ is the virtual potential temperature (K), calculated from the sonic anemometer virtual temperature data $T_v$ and the measured pressure $p$ as $\theta_v = T_v \left(\frac{p_0}{p}\right)^{R/c_p}$ with $p_0 = 1000$ hPa, and $R/c_p \approx 0.286$; $k = 0.4$ is the von Kármán constant; $g = 9.81$ m s$^{-2}$ is the acceleration due to gravity; $u_* = (\overline{u'w'}^2 + \overline{v'w'}^2)^{1/4}$ is the friction velocity (m s$^{-1}$); and $\overline{w'\theta'_v}$ is the kinematic sensible heat flux (W m$^{-2}$).

Reynolds decomposition for turbulent flows is applied to separate the average and fluctuating parts of the relevant quantities. The average time period used to compute the Reynolds decomposition must be much longer than any turbulence time scale, but much shorter than the time-scale for mean flow unsteadiness. For this purpose, it has been fixed to 30 minutes, a typical time range used to compute turbulent averages for atmospheric boundary layer phenomena (De Franceschi and Zardi, 2003; De Franceschi et al., 2009; Babić et al., 2012).

As to atmospheric stability, we consider neutral atmosphere for $L \leq -500$ m and $L > 500$ m; unstable conditions for $-500$ m $< L \leq 0$ m; and stable conditions for $0$ m $< L \leq 500$ m (Muñoz-Esparza et al., 2012).

## 3 Wake characterization algorithm for multiple wakes

The line-of-sight velocity ($u_{LOS}$) measured by the WINCDUBE 200S scanning lidar (Figure 4) during the horizontal (PPI) scans can be analyzed to determine wake characteristics and how they evolve in space as the wakes propagate. Aitken et al. (2014a) proposed a wake detection algorithm and applied it to characterize the wake from a single turbine, and later expanded it to treat nacelle-based lidar measurements (Aitken and Lundquist, 2014). Here we expand the same algorithm to characterize wakes from a row of four turbines.

### 3.1 Data pre-processing

First, a threshold is imposed to the carrier-to-noise ratio (CNR), which represents the strength of the backscattered signal compared to background noise (values closer to 0 dB indicate a stronger signal relative to the noise): all measurements with carrier-to-noise ratio $CNR < -27$ dB are discarded from further analysis (Vanderwende et al., 2015). Measurements with a lower CNR often had unrealistically high ($> 15\,\mathrm{m\,s^{-1}}$) values of radial velocity; and this threshold value is comparable with choices in other studies (Cariou et al., 2011; Bastine et al., 2015; Debnath et al., 2016). Moreover, in each scan, line-of-sight velocity data which are not included in the interval $(\mu - 3\hat{\sigma}, \mu + 3\hat{\sigma})$, where $\mu$ is the average of the data, are removed from the analysis. The standard deviation $\hat{\sigma}$ is evaluated according to the median absolute deviation (MAD), assuming normally distributed data: $\hat{\sigma} = 1.4826\,MAD$, where $MAD = \mathrm{median}\,(|u_{LOS,i} - \mathrm{median}(u_{LOS})|)$. In the remaining part of the wake detection algorithm, measurements will be weighted according to the inverse of the square of the radial wind speed dispersion, which is a measurement of the standard deviation of the back-scattered signal of the lidar and thus an indicator of the uncertainty of values (Vanderwende et al., 2015).

### 3.2 Wake detection

To implement the wake detection, measurements of line-of-sight velocity $u_{LOS}$ at each range gate in each PPI scan are fitted to two different models: the first is for ambient flow conditions without wakes, the second represents each of the four wakes as a Gaussian function subtracted from uniform ambient flow (Tennekes and Lumley, 1972; Troldborg et al., 2007; Chamorro and Porté-Agel, 2009; Gaumond et al., 2014). Ambient wind speed is modeled with uniform speed $u$ and direction $\phi$, as shown in Figure 4. At a fixed elevation angle, the line-of-sight velocity $u_{LOS}$ can be related to the assumed uniform ambient wind speed $u$ with a simple geometric transformation involving horizontal wind direction $\phi$ and the lidar azimuth angle $\alpha$, namely:

$$u_{LOS} = u \cdot \cos(\alpha - \phi) \tag{2}$$

where both $\alpha$ and $\phi$ are $> 0$ for clockwise rotations from North. The azimuth angle $\alpha$ can be related to the range gate $r$ and the transverse coordinate $y = r \cdot \cos(\theta) \cdot \sin(\alpha)$, yielding:

$$u_{LOS}(y, r) = u \cdot \left( \frac{\sqrt{r^2 - y^2}}{r} \cos\phi + \frac{y}{r} \sin\phi \right) \tag{3}$$

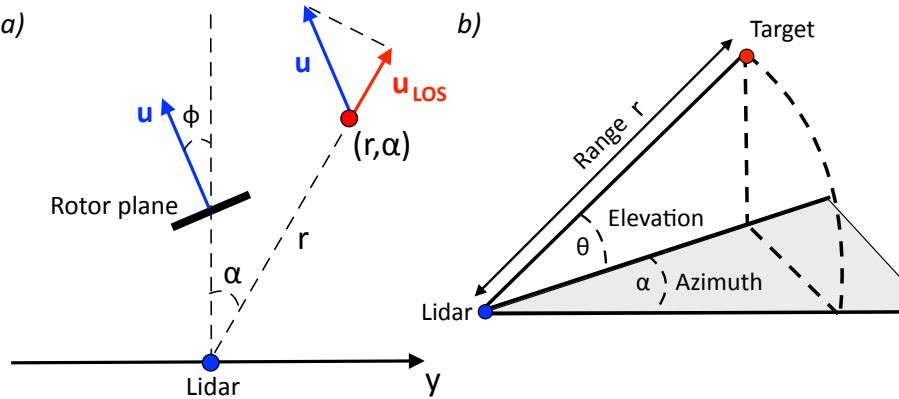

**Figure 4.** a) Plan view of the coordinate system for scanning lidar PPI scans. b) 3-D sketch of the main geometric quantities relevant in a lidar scan.

which represents the first model for $u_{LOS}$ applied in the wake detection algorithm. In this case, the ambient flow wind speed $u$ and the ambient wind direction $\phi$ are the fitting parameters of the model. This "no wake" fit is the same as in Aitken et al. (2014a).

The second implemented model represents each wake from the four turbines in the row as a Gaussian function (Tennekes and Lumley, 1972) subtracted from uniform ambient flow $u$:

$$u_{LOS}(y,r) = \left\{ u - \sum_{i=1}^{4} a_i \exp\left[ -\frac{(y-y_i)^2}{2s_{wi}^2} \right] \right\} \cdot$$
$$\cdot \left( \frac{\sqrt{r^2 - y^2}}{r} \cos\phi + \frac{y}{r} \sin\phi \right) \tag{4}$$

This second model has 14 fitting parameters: the ambient wind direction $\phi$, the ambient wind speed $u$, the amplitudes of the Gaussian functions (i.e. the wake velocity deficit amplitudes) $a_i$, the four transverse coordinates of wake centers $y_i$, and four parameters controlling the widths of the wakes $s_{wi}$. Note that each of the four wakes is modeled with its own parameters, permitting variable characteristics between the wakes. The amplitude $a_i$ can be 0, for the trivial case of no wake.

Nonlinear regression (least squares) is applied with the two different models specified above. In the fitting procedure, the transverse coordinate $y$ is used as the independent variable, while the measured line-of-sight velocity $u_{LOS}$ is the dependent variable; moreover, the dispersion of measured $u_{LOS}$ is used as weights for the data. In setting the first-guess values for the parameters, physical limits are set: the velocity deficit amplitudes must be $\geq 0$ but lower than the uniform ambient flow wind speed $u$; the locations of the centers of the wakes must be included in the range of transverse coordinates $y$ in each scan; the width of the wakes must be $> 0$.

The best estimates for the parameters of the two models are found. An extra sum-of-squares $F$ test is applied to determine if the second model, which is naturally suited to better fit data considering its higher number of parameters, is *significantly* better than first model in fitting the data. A threshold $p$ value is set to 0.05; if the calculated $p$ value is less than this threshold, then the

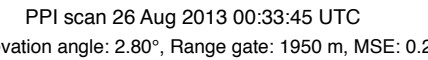

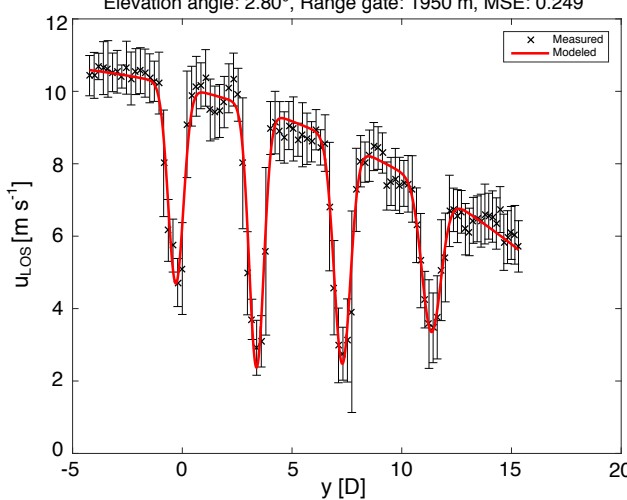

**Figure 5.** Example of line-of-sight velocity data measured by the scanning lidar at a specific range gate during a PPI scan. Error bars are the dispersion of line-of-sight velocity measurements. The red line represents the fit performed by the wake characterization algorithm.

second model is considered capable to *significantly* better represent the data, and thus it is selected (Kleinbaum et al., 2013). Figure 5 shows an example of line-of-sight velocity measurements at a single range gate in a PPI scan (with error bars representing the dispersion of the measurements). The red continuous line is the fit performed by the wake characterization algorithm.

### 3.3 Model acceptance criteria

The quality controls implemented in the first steps of the wake detection algorithm (limits to CNR, MAD method to discard outliers, physical limits to the values of the fit parameters) assure a very good quality of the fits, measured in terms of Pearson correlation coefficient and mean squared error. However, some other quality-control steps are applied, to solve possible issues related with the complexity of the expansion of the algorithm to detect multiple wakes.

At the smallest range gates, depending on a given wind direction, not all the four wakes from the studied row of four turbines may be included in the lidar scan because of the limited range of azimuths of each scan. In these situations, the application of the second model, which aims at fitting $u_{LOS}$ with four wakes, may result in the detection of some spurious wakes besides the actual (but less than four) real wakes seen in the PPI scan. These spurious wakes are typically detected where sudden - but very limited - natural changes in the line-of-sight velocity occur. To solve this false detection, fitted wakes with non-realistic velocity deficits (smaller than half of the minimum velocity deficits of the other wakes detected at the same range gate) and/or widths (smaller than $0.1D$ or bigger than $1/4$ of the whole transverse range) are excluded from the results.

Another issue is related to the possibility for the algorithm to detect wakes with a double-peaked shape - typical of near-wake (Magnusson, 1999), or arising from the interference of an obstacle with the laser beam of the lidar - as two separate wakes.

The algorithm detects a double-peaked wake when the transverse positions of the centers of two adjacent wakes are closer than half of the width of the largest wake. When these double-peaked wakes are detected, the algorithm can instead consider a single-peaked wake with a velocity deficit amplitude which is the average of the two detected velocity deficit amplitudes, a wake center located at the average $y$ between the two detected peaks, and a wake width determined adding a half-width ($= 2s_{wi}$) to each external edge of the two detected peaks.

Besides these quality-control steps, the algorithm can re-order the remaining wake parameters, to associate them to the correct physical wake from the considered row of four turbines. This procedure is dependent on wind direction, that determines in which order the four wakes are excluded from the scan area of the lidar at lower range gates.

The wake characteristics database, resulting as output of the application of the wake detection algorithm (which is publicly available at https://github.com/nicolabodini/CWEX13), is then used to study how wake characteristics evolve in 3-dimensional space.

For each detected wake, the velocity deficit is calculated as the ratio between the velocity deficit amplitude $a_i$ and the ambient flow wind speed $u$ (estimated from our algorithm at each performed fit at each range gate and elevation) (Vermeer et al., 2003):

$$VD_i = \frac{u - u_{wake}}{u} \cdot 100 = \frac{a_i}{u} \cdot 100 \qquad (5)$$

The wake width has been defined in different ways in the literature; here we calculate it as in Hansen et al. (2012):

$$w_i = 4 \cdot s_{wi} \qquad (6)$$

which is equivalent to the 95% confidence interval of the Gaussian velocity deficit profile.

The wake centerline will be studied considering the temporal evolution of the planar coordinates of the center of each wake, i.e. the peak of the velocity deficit.

As final quality-control steps, the MAD method is applied again to discard wake characteristics which do not lie within three standard deviations of the mean characteristic at each range gate for each whole night (Aitken et al., 2014a); moreover, only fits with Pearson correlation coefficient ($corr(u_{LOS}, \hat{u}_{LOS}; g) = cov(u_{LOS}, \hat{u}_{LOS}; g)/\sqrt{cov(u_{LOS}, u_{LOS}; g)cov(\hat{u}_{LOS}, \hat{u}_{LOS}; g)}$, where $g$ represents the data weights) larger than 0.9 and mean squared error ($MSE = \frac{1}{\sum_{i=1}^{n} g_i} \sum_{i=1}^{n} g_i(\hat{u}_{LOS,i} - u_{LOS,i})^2$) lower than 0.5 are included in the final analysis of the results.

## 4    Results

Once all the fits are completed, and the wakes fully characterized, it is possible to study how the wake characteristics vary in space for the four studied turbines.

### 4.1    Frequency of wake detection and atmospheric stability

Atmospheric stability has a major impact (Churchfield et al., 2012; Iungo et al., 2013) on wind turbine wake evolution and wind farm performance: wakes in stable conditions persist for long distances downwind, while during unstable conditions the

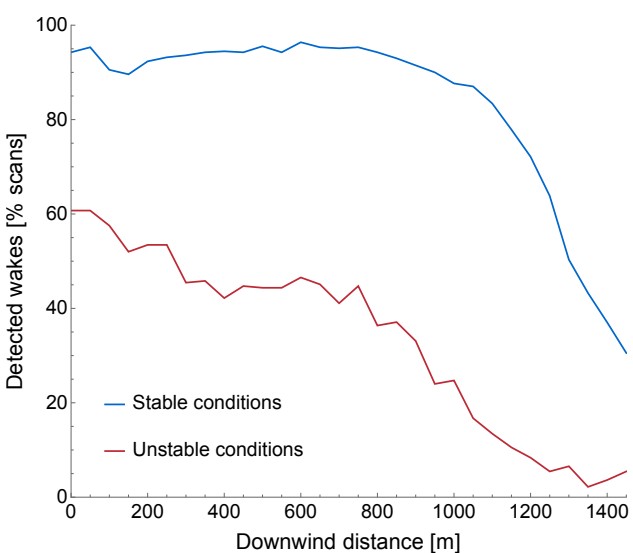

**Figure 6.** Percentage of wakes which are detected by the characterization algorithm *vs* downwind distance, for stable and unstable conditions of the atmosphere (neutral conditions were detected only for very short periods, and are not included here). Results from PPI scans performed on 23 and 26 August 2013.

enhanced turbulent mixing erodes the wakes more quickly.

To get a quantitative measurement of this effect, Figure 6 shows the percentage of scans where wakes were detected by the algorithm, at each range gate, for different stability conditions of the atmosphere (measured in terms of the Obukhov length) during all the 438/576 scans (at all the considered elevation angles) performed on 23/26 August 2013. The plot clearly shows how wakes can easily be detected in stable conditions, while during unstable conditions the algorithm is not capable of properly detecting wakes at least 40% of the times. Moreover, wakes erode more quickly during unstable conditions, with the degradation becoming more intense approximately at 700 m ($\sim 8.5D$) downwind of the wind turbines, while under stable conditions wakes are detected in most of the scans up to approximately 1000 m ($\sim 12D$) downwind of the turbines.

All the results presented in the next paragraphs focus on stable conditions of the atmosphere.

## 4.2 Velocity deficit results

Wind speed reduction in the wake region, measured in terms of velocity deficit, is the most distinct wake effect. Figure 7 shows contour plots of velocity deficits for a wake from an outer turbine (panel $a$) and a wake from an inner turbine (panel $b$), computed using the results of the wake detection algorithm from PPI scans performed at six different elevation angles from 05:31 to 05:42 UTC (from 00:31 to 00:42 LDT), 26 August 2013. The horizontal axis shows the downwind distance from the turbines, expressed in terms of rotor diameters $D$, with $D = 80$ m. The plot clearly shows that the velocity deficit decreases with downwind distance, and the wake of the outer turbine exhibits smaller velocity deficits compared to the wake of the inner turbine.

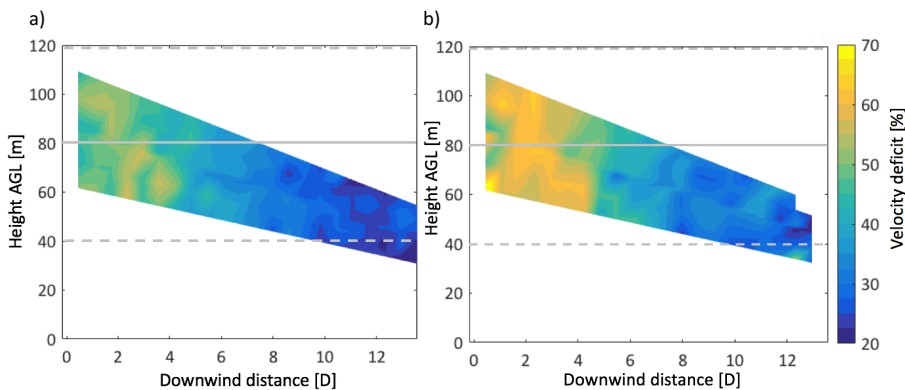

**Figure 7.** Velocity deficit $vs$ downwind distance, at different vertical positions, for wakes from a) an outer turbine and b) an inner turbine. Gray horizontal dashed lines represent the vertical limits of the rotor disk of the turbines; the horizontal continuous gray line shows the hub height of the turbines. Data collected from 5:31 to 5:42 UTC (from 00:31 to 00:42 LDT), 26 August 2013, from a succession of six PPI scans performed at six different elevation angles.

To understand if the results are systematic, Figure 8 shows velocity deficit versus downwind distance from the turbines, calculated from the 276 (242) PPI scans, at all the elevation angles, performed during the whole night - stable conditions - of 26 (23) August 2013. The continuous lines show the median values of velocity deficit, and the shaded area represents the standard deviation of the data. As expected, velocity deficit decreases with downwind distance, since the speed reduction in the wake
tends to become smaller due to the entrainment of free-stream surrounding air. The plot also confirms that wakes from outer turbines (number 1 and number 4), have lower velocity deficits than the wakes from inner turbines (number 2 and number 3), for relatively small downwind distances, with a difference up to 15%. The presence of outer turbines seems to reduce the effectiveness of lateral entrainment of faster air to recover wind conditions in the inner wake regions of the wind farm. These results are comparable for both the considered nights: different wind directions do not seem to affect them.

**4.3   Wake width results**

The widths of the wakes also change with downwind distance (expressed in terms of rotor diameters $D$, Figure 9). Panel $a)$ shows results from the stable conditions of the night of 26 August 2013, while panel $b)$ shows results for the stable conditions of the night of 23 August 2013. In both cases, for all the four turbines, the wake widths increase moving away from the turbine, exceeding $2\,D$ after a downwind distance of 8-10 $D$.
However, if we focus on the single wakes, we can see how different wind directions (southwesterly during 26 August, southeasterly during 23 August) can affect the ability of the scanning lidar to measure line-of-sight velocity and, thus, detect this characteristic of the wakes. By comparing the two plots in panels $a)$ and $b)$, it is clear that the scanning lidar systematically identifies as the widest the wake which, at a given downwind distance, is the most perpendicular to the laser beam (i.e. the last one the laser beam meets: turbine 4, at the right edge of the row, for southwesterly wind; turbine 1, at the left edge of the

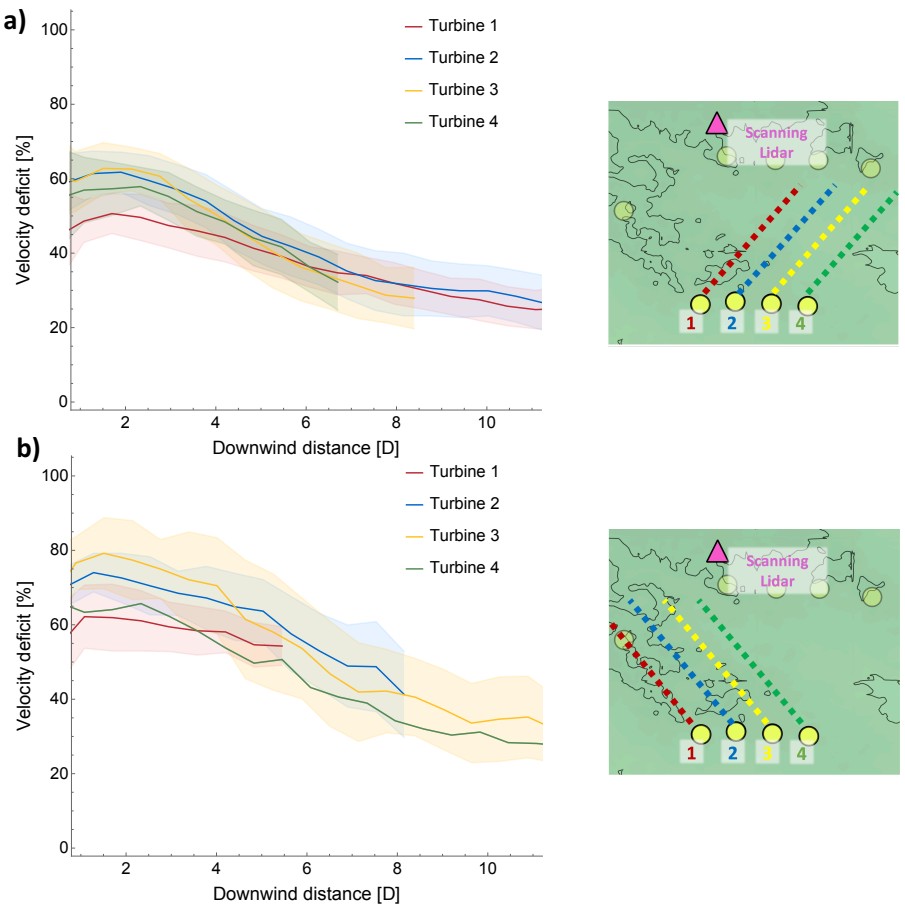

**Figure 8.** Velocity deficit $vs$ downwind distance, for the four wakes of the studied row of turbines. Continuous lines represent the median values calculated from the PPI scans performed at all the considered elevation angles during the night (stable conditions) of 26 (panel a) and 23 (panel b) August 2013; shaded areas show $\pm$ one standard deviation of the data.

row, for southeasterly wind). Then the detected width of the wakes progressively decreases moving to the wakes from adjacent turbines.

Panel $c$) in Figure 9 aggregates results from 23 and 26 August, and it shows how wake width changes with downwind distance considering the single turbines from the closest to the furthest from the scanning lidar, depending on the particular wind direc-

5  tion, as shown in the right schemes in the panel. The plot confirms the systematic dependence of the detected wake widths on the relative position between the wake and the scanning lidar.

This result is due to the relationship between the viewing angle and the aspect ratio of the lidar retrieval "pixels", which are related to the relatively long range gate (50 m) and relatively narrow azimuthal resolution (0.5°). As qualitatively shown in the schematic of Figure 10, the scanning lidar measures the line-of-sight velocity in narrow pencil-shaped "pixels". With this

10  geometry, if the wind direction - and thus the wake - is aligned with the line-of-sight from the lidar, the wake width can be

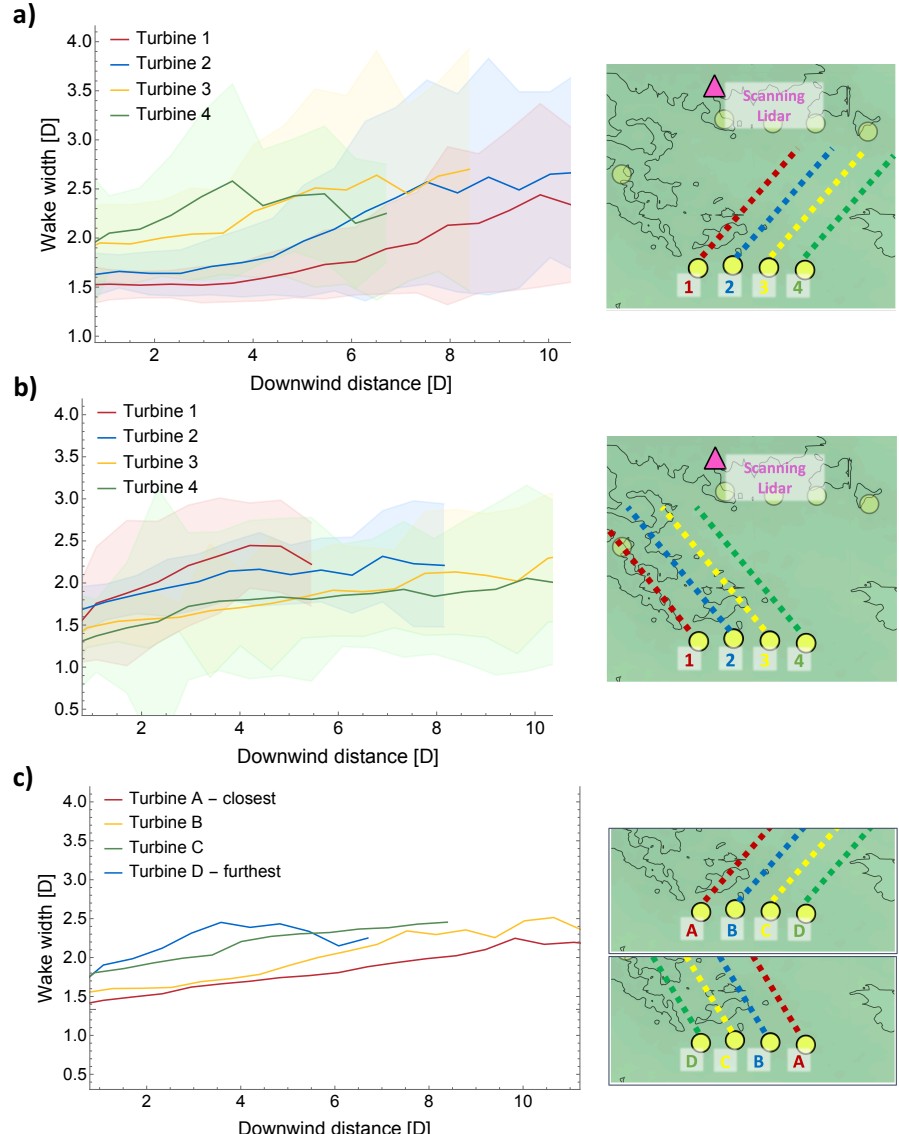

**Figure 9.** Wake width $vs$ downwind distance from the turbines, for the wakes of the four turbines in the studied row, from PPI scans performed at all the six considered elevation angles. Continuous lines represent median values; shaded areas show $\pm$ one standard deviation of the data. a) Data from the night (stable conditions) of 26 August 2013, with southwesterly wind conditions. b) Data from the night (stable conditions) of 23 August 2013, with southeasterly wind conditions. c) Aggregated plot, average of data from the nights of 26 and 23 August, considering the single turbines with reference to their relative distance from the scanning lidar.

assessed with high precision due to the high azimuthal resolution in each pencil-shaped area (panel a). However, if the wind direction - and thus the wake - is not aligned with the line-of-sight from the lidar (panel b), then the same wake will be measured as generally wider, since the retrieval of the wake width is now affected by the relatively coarse radial resolution of the

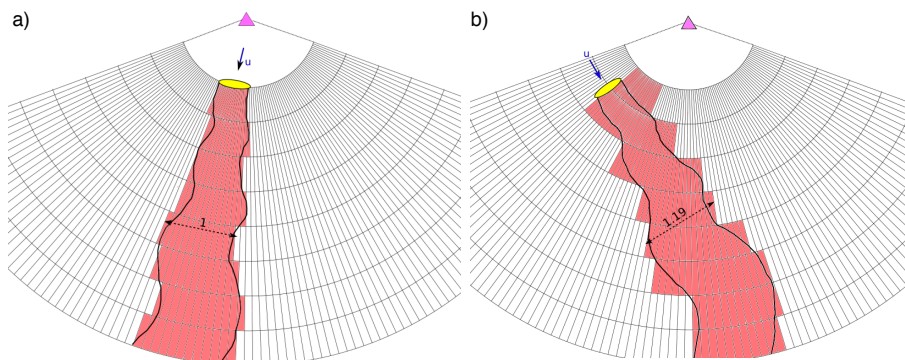

**Figure 10.** Qualitative sketch of the dependence of detected wake width on the orientation of the coordinate grid used by a scanning lidar (purple triangle) as a function of the wind direction. Panel (a) shows the case of a wake aligned with the line-of-sight from the scanning lidar (wind direction shown by the blue arrow), while panel (b) shows the case of a wake not aligned with the line-of-sight from the lidar. The dashed arrow highlights the difference in the detected wake width for the two cases, at fixed downwind distance from the turbine (yellow ellipse).

lidar coordinate grid. In the schematic diagram shown in Figure 10, at an arbitrary fixed downwind distance from the turbine, the (same) wake would be detected as 19% larger when it is not aligned with the line-of-sight from the scanning lidar. This result becomes more evident when the laser beam is more perpendicular to the wake. This result is due to the aspect ratio of the lidar "pixels," and thus would affect other wake characterization approaches relying on instruments not co-located with the turbine - such as in Banta et al. (2015); Aitken et al. (2014a) - but would not affect nacelle-mounted wake measurements, such as in Bingöl et al. (2010); Aitken and Lundquist (2014) as nacelle-mounted wake measurements are usually aligned with the wake, unless the wake is intentionally yawed (Fleming et al., 2016; Trujillo et al., 2016).

## 4.4 Wake centerline results

PPI scans at multiple elevation angles provide insight into the 3-dimensional structure of wind turbine wakes. Different conditions at different vertical levels have a considerable impact on the wake centerline, i.e. the change of the position of the wake center downwind of the turbine. Figure 11 shows a plot of the median position of the centers for the wakes of the two turbines located at the west edge of the considered row of four turbines, for the 2:30 - 3:30 UTC (21:30-22:30 LDT) time period during the night of 26 August 2013 (southwesterly wind, panel $a$) and for the two turbines located at the east edge of the considered row of four turbines for the 9:30 - 10:30 UTC (4:30-5:30 LDT) time period during the night of 23 August 2013 (southeasterly wind, panel $b$). Dashed lines represent the median value for the wake centerlines; continuous lines show the results for data points with different vertical heights: light colors show results for points with a vertical height between 35 m and 55 m, dark colors refer to measurements taken above 75 m AGL (these levels were chosen to create bins at low and high heights, compared to the vertical dimension of the turbines, with approximately the same number of vertical positions where the lidar measurements were taken, as shown in Figure 2). A clear change in the position of the wake centers is detected between low

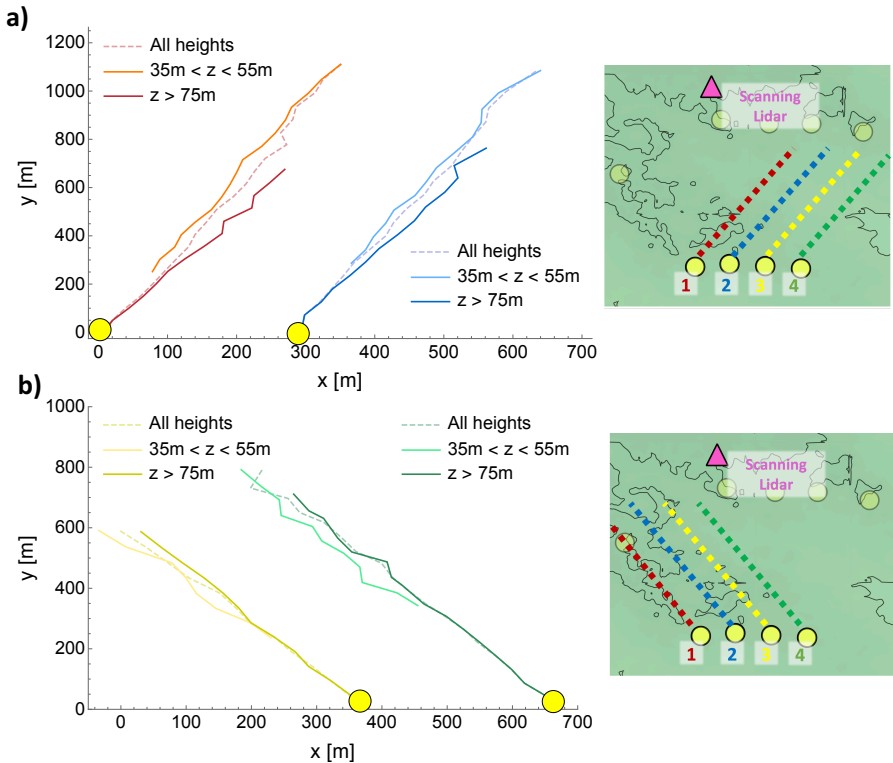

**Figure 11.** a) Wake centerlines for an outer (on the left) and inner (on the right) turbine, from the PPI scans measurements during the 2:30-3:30 UTC (21:30-22:30 LDT) time period, 26 August 2013. b) Wake centerlines for an inner (on the left) and outer (on the right) turbine, data from the 9:30-10:30 UTC (4:30-5:30 LDT) time period, 23 August 2013. Dashed lines represent the median values of wake centerlines, while the continuous lines show results at different vertical levels: for each wake, light colors refer to measurements between 35 m and 55 m AGL, while darker colors represent data points with a vertical height greater than 75 m. Yellow dots show the position of wind turbines.

and high vertical levels. This stretching is independent of wind direction: the change can be seen for both southwesterly (panel $a$, 26 August) and southeasterly (panel $b$, 23 August) wind conditions.

This change of the wake centerline with vertical height causes a stretching of the vertical structure of the wakes: the velocity deficit structure of a turbine wake, whose stream-wise velocity deficit is traditionally considered as a 3-D Gaussian in a

5 cross-stream plan (Figure 12 - $a$), should instead be represented - when this vertical stretching occurs - by a rotated ellipsoid (Figure 12 - $b$), as already observed in both field measurements (Högström et al., 1988; Magnusson and Smedman, 1994) and large-eddy simulations (Lundquist et al., 2015; Vollmer et al., 2016).

### 4.4.1 Relationship between ambient veer and wake centerline

The vertical stretching of wake structure occurs because of the wind veer, the clockwise change of wind direction with height

10 that occurs in strongly stable conditions, like those of 23 and 26 August 2013. To get a deeper insight on the relationship

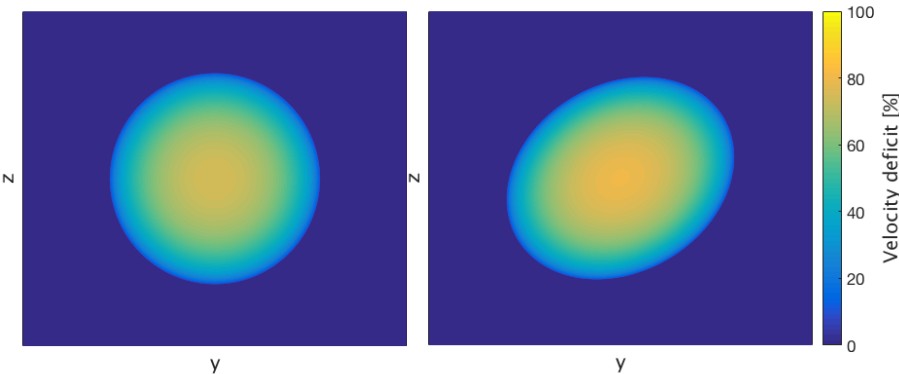

**Figure 12.** Qualitative cross-stream slices of stream-wise velocity deficit. The perspective is looking downwind. On the left, a graphic representation of the classical Gaussian shape of the magnitude of wake velocity deficit. On the right, an ellipsoid which represents the vertical stretching of the 3-D structure of a turbine wake as a consequence of wind veer.

between ambient wind veer and vertical changes of the wake centerlines, we analyze several 30-minute periods (each corresponding to two subsequent sequences of six PPI scans at the six different elevation angles) during the nights of 23 August 2013 (southeasterly wind) and 26 August 2013 (southwesterly wind). For each considered time frame, the wind veer is calculated as the average difference between the wind direction at $100$ m and $40$ m, as measured by the vertical profiling lidar located to measure upwind conditions (WC-1 in Figure 1). The two vertical heights are chosen as representative of the two different vertical levels considered when assessing the changes of wake centerlines ($35$ m $< z < 55$ m and $z > 75$ m). Moreover, the wake centerlines at different vertical levels have been fitted with straight lines, and the angle between the lines which approximate the wake centerlines at $35$ m $< z < 55$ m and $z > 75$ m is calculated and then considered as the angular difference between the wake centerlines at different vertical levels. Figure 13 shows how angular difference between wake centerlines at different vertical levels compares to ambient wind veer between $100$ m and $40$ m, for all the available time frames during the nights of 23 and 26 August 2013, for wakes from an inner and an outer turbine in the considered row. The results show that, although the angular change in the wake centerline at different vertical levels is systematically detected, the wind veer is always much larger than the actual angular difference between the wake centerlines at the different vertical levels: the change of the positions of the wake centers is related to, but not completely determined by, the wind veer. Moreover, as suggested by the linear regression fits, wakes from outer turbines often present a larger angular difference in wake centerlines compared to wakes from inner turbines, though with variability for different veer values that motivates further study.

A possible physical explanation for this phenomenon can be detected in the interaction between wake rotation due to rotating blades and wind veer. The blades of the wind turbines in CWEX-13 wind farm rotate clockwise and so the downwind wakes rotates counter-clockwise (Burton et al., 2001).The wake can thus be considered as a sort of plume with its own momentum and rotation, which interacts with the ambient veer, that in turns tends to rotate the wake in the opposite direction (in the Northern hemisphere), thus causing a reduction in the global wake vertical stretching than would be present if only the ambient veer

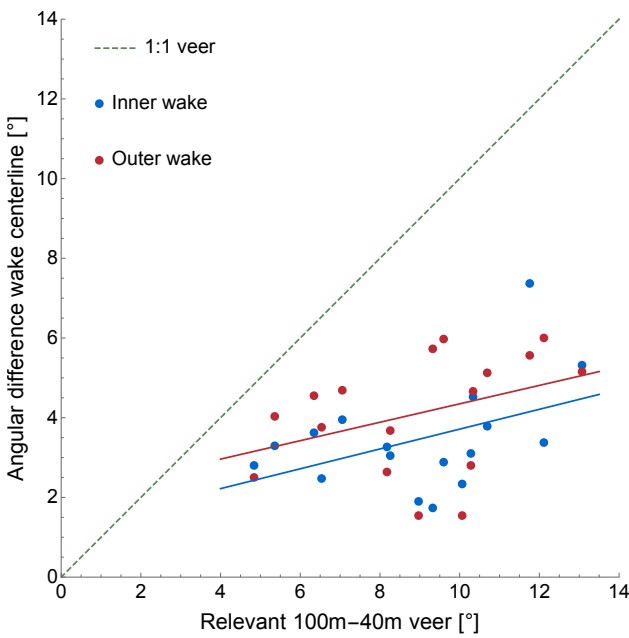

**Figure 13.** Angular difference between wake centerlines at different vertical heights ($35$ m $< z < 55$ m and $z > 75$ m) *vs* wind veer between 100 m and 40 m. Data for an inner and an outer wake in the considered row of four turbines, for several 30-minutes time frames during the nights of 23 and 26 August 2013. Continuous lines are the linear regressions from the data (red best-fit: $2.04 + 0.23x$, blue best-fit: $1.23 + 0.25x$), for the outer and the inner wake. Dashed line highlights the not-reached equality between ambient veer and wake centerlines angular difference.

affected the wake. Inner wakes seem to be less subject to the effect of ambient wind veer, as if the presence of outer turbines reduces the ability of ambient wind characteristics to reach and impact inner regions of the wind farms.

## 5    Conclusions

Wakes from a row of four turbines have been characterized using line-of-sight wind speed measurements from PPI scans
5    performed by a scanning lidar. Data were collected in late summer 2013 during the CWEX-13 field campaign (Lundquist et al., 2014), in a wind farm in a flat region in central Iowa. The wake characterization algorithm proposed by Aitken et al. (2014a) has been extended to assess wakes from multiple turbines.

Wakes erode quickly during unstable conditions of the atmosphere, and they can in fact be detected here primarily in stable conditions in this dataset. The velocity deficit in the wakes decreases with downwind distance from the turbines, and it is lower
10    for wakes from outer turbines in the studied row. The width of the wakes increases with downwind distance, with systematic differences in the ability of the scanning lidar to detect the width of the wake according to the component of the direction the wakes perpendicular to the direction of laser beam of the scanning lidar. Wake centerlines change at different vertical

levels as a consequence of the ambient wind veer, causing a stretching of the vertical structure of the wakes. Although the field measurements of Högström et al. (1988); Magnusson and Smedman (1994) demonstrated that turbine wakes stretch into ellipses during stable conditions, for the first time we have quantified the effect of ambient wind veer on the stretching of wakes. In fact, the angular change in the wake centerlines at different heights is systematically much lower (a half or less) than

the wind veer registered at the same heights. Moreover, this angular change of the wake centerlines at different vertical levels is found to be usually greater for wakes from outer turbines. This wake stretching, due to wind veer, is not only seen in these field measurements but also emerges in the stably-stratified simulations of Aitken et al. (2014b); Bhaganagar and Debnath (2015); Lundquist et al. (2015); Vollmer et al. (2016); Abkar et al. (2016). As more three-dimensional measurements of wakes become available due to the use of scanning lidar and scanning radar, a more solid representation of wind turbine wakes can

be assessed.

These results can become critically important to assess and improve large-eddy simulations of wakes as well as to suggest improvements to mesoscale parametrizations (Fitch et al., 2012, 2013; Jiménez et al., 2015; Lee and Lundquist, 2017) to account for subgridscale wake interactions. Moreover, wind energy companies can also benefit from our results in trying to enhance the quality of low-order wake models currently used for wind resource assessment, wind farm layout optimization and

wind farm control techiniques, with the final goal of an improvement of wind energy production efficiency.

## 6   Code availability

The Matlab code of the wake characterization algorithm is publicly available at https://github.com/nicolabodini/CWEX13.

## 7   Data availability

CWEX-13 data will be publicly-available at the Dept. of Energy Atmosphere to electrons archive at https://a2e.energy.gov/projects.

*Acknowledgements.* This work was partially supported by the National Renewable Energy Laboratory under APUP UGA-0-41026-22 and by the National Science Foundation grant BCS-1413980 (Coupled Human Natural Systems). Nicola Bodini was partiallly supported by a grant from the University of Trento for a visit to the University of Colorado Boulder in summer 2016.

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
