# Peer review of "Three-Dimensional Structure of Wind Turbine Wakes as Measured by Scanning Lidar"

_Atmospheric Measurement Techniques, 2017_

## Referee Comment (RC1) · Anonymous Referee #1 · 5 May 2017

SUMMARY: This paper discusses the analyses of wind turbine wake structure considering varying atmospheric stability regimes using measurements from a single scanning lidar; representing an extension of the work of Aiken and Lundquist (JTECH, 2014). Frequency in wake detection, wake velocity deficits, wake width, and wake centerline results are presented. I found the discussion related to the wake stretching as a result of vertically veering wind direction to particularly interesting. The impact of atmospheric stability on wake behavior is a very important aspect to understanding wind plant performance as a whole, and as this paper shows there are significant and consistent (e.g. diurnal) changes in wake behavior as a result of the background stability. I believe the results of this paper are a meaningful contribution and worthy of publication, but I do have a few comments/questions requesting clarity for how the measurements are used to construct the analyses and what the downstream implications are for the presented results.

MAJOR COMMENTS/QUESTIONS: While the title infers a 3D construction of the wakes for analysis, in reality, all of the presented wake statistics are only assessed from individual 2D planes. Since multiple elevation tilts are used, I would have liked to see the author construct fully integrated 3D volumes of data by interpolating the polar data from a given elevation series (by elevation series I mean a single collection of the six elevation tilts between 1.5-2.8 degrees) into a 3D Cartesian grid. Otherwise, when assessing wake deficit and width, how can we be certain that any given 2D plane is a complete representation of the absolute downwind wake deficit or wake width since that plane only represents a horizontal 2D "slice" somewhere through the wake? Since deficits and widths were computed separately from the individual PPI elevation planes within the same elevation series, this would mean multiple wake deficit/width calculations for the same downwind distance exist within the same elevation series, but inherently represent different vertical locations within the wake. Or maybe the generated wake statistics for one elevation series are binned by range regardless of their height? Additional discussion on how the statistics from individual tilts within a single elevation series are merged (or not) would provide for a better understanding of the bulk wake statistics (e.g. deficit and width) and how representative the statistics are of the character of the full wake at any given downwind distance. I apologize if I have misinterpreted the analysis, but I believe there needs to be further clarification of how the 2D planes of PPI data are used to generate the wake statistics, and what inherent assumptions/limitations are associated with the methods. I would also encourage the author, though not required, to consider analyzing the data in a true 3D framework when constructing the presented wake statistics.

MINOR COMMENTS/QUESTIONS: 1. P4, L8: What is the range from the WINDCUBE to the four turbines of interest? The distances can be inferred from Figure 2, but the numbers would be useful in the text.

2. P 4, L 14: What is the scan speed of the lidar? What range of azimuths are scanned?

3. P4, L 19: How long does it take the lidar to scan the series of six elevation tilts?

4. P9, L 11: This comment relates to the Major Comments/Questions section above. How is the ambient flow wind speed defined on the 2D PPI plane? Since the PPI plane is slanted, if the ambient flow wind speed is determined upstream of a given turbine, can you comment on the impact of using this value for constructing wake deficits downstream of the turbine, but at lower heights due to the slanted PPI plane? Is the comparison being made at different heights because the PPI plane is sloped? If so, what are the implications on the wake deficit calculations?

5. P9, L30: How many scans are used to generate the statistics in Figure 5? Are all of the 438/576 scans performed on 23 Aug/26 Aug considered? Are all elevation tilts considered?

6. P10, L13: Does this statement imply that a single elevation series of six scans takes 11 minutes to complete? If so, that may answer Question 3 above.

7. P10, L18: While the difference in wake deficit between the inner and outer wakes is shown, can the author comment on the difference in wake deficit between the outer turbines, as that difference is actually more substantial? Which line is for which turbine? Assuming the wind direction is relatively consistent throughout (maybe a bad assumption), can a composite PPI image be provided corresponding with this period to highlight if there are any features in the flow (e.g. a turbine row edge effect) that could be contributing?

8. P12, L2: While the author states comparable results occur between the 23-August and 26-August cases, do the red lines, for example, flip with the different wind direction, again inferring some type of turbine row edge effect?

9. P12, L18: Could an example PPI image or composite be included to visually highlight the wind direction dependence on wake width detection being discussed here?

10. P12, Section 4.4: This comment relates to the Major Comments/Questions section above. Perhaps consider constructing a 3D volume of interpolated information, as opposed to compositing two horizontal planes for comparison. The change in wake centerline with height, supported by the presented measurements, is a really neat result of this study. The actual shape of the ellipsoid could potentially be better described (and compared to previous measurements) if the data were constructed in a 3D framework.

11. P12, L28: Why were the data separated between 55-75 m and >75 m, especially given hub height is at 80 m and data exist below 55 m within the rotor sweep. Why was the lower region bound vertically by 20 m but the upper region allowed to be larger? A quick comment on why this method was chosen could be beneficial.

MINOR EDITS: 1. P2, L16: "four" instead of 4.

2. P4, L15: The period at the start of this line should be at the end of the previous line.

3. P4, L23: "six" instead of 6.

4. P6, L14: Insert a period after (MAD).

5. P7, L11: Insert the word "the" between "as independent".

6. P9, L19: MAD is already defined on P6.

---

## Referee Comment (RC2) · N. Gayle Nygaard (Referee) · 8 May 2017

Summary: this paper presents single lidar measurements of multiple wind turbine wakes. A wind field reconstruction model is used to derive wake characteristics from the line of sight wind speeds. The presented model is an extension to multiple wakes of a previous model developed by the same research group to characterize a single wake. Results are presented for the decay of the velocity deficit and the increase of the wake width with downstream distance. The wake centerline is found to shift orientation with height, indicating stretching of the wake profile in veered flow. The results are interesting and add to the developing picture of wakes in wind farms.

The method is sound, but could have been better explained, especially with regards to the vertical dimension and the influence of multiple elevation tilts (see specific comments below). The authors find some differences between the outer and the inner turbines in a row, but no explanation for these differences is offered.

The paper is definitely worthy of publication, but I suggest the authors consider the questions and suggestions below to improve the presentation of the methodology and the results.

Questions and suggestions:

In the model of Eq. (4) there is no reference to the vertical dimension or the elevation angle. Is it supposed to be applied at a fixed elevation angle? It is unclear how the vertical structure of the wakes is considered. At a fixed elevation angle the laser beam will probe at increasing height with increasing range. Were multiple elevation tilts combined in figures 7 and 8 to account for this?

Consider adding a reference to Wang and Barthelmie - Journal of Physics: Conference Series 625 (2015) 012017 - Wind turbine wake detection with a single Doppler wind lidar. This has a similar wake wind field reconstruction method.

P1, L23: The reduction in power for turbines in wake can exceed the 40% mentioned as the upper limit. As is well known, it is very sensitive to wind direction, being largest when the wind is aligned with rows of turbines. Also the observed maximum reduction depends on the size of the wind direction sector over which the data are averaged. But even for a 30 degree sector Nygaard 2014 found reductions of up to 60% for certain conditions. The total wake loss considering all wind directions and wind speeds is typically less than 20%. My point is that the 40% mentioned in the text is a meaningless number without further context. It only applies for certain wind directions and for averaging over a sector of a certain size. I invite you to make the context of this number clearer or to consider, if a specific value is needed.

P2, L11: the appropriate reference for the Jensen model is N. O. Jensen, A note on wind generator interaction, Risø-M-2411 (1983). The reference you have to Jensen

1984 was new to me, so thank you for pointing it out. However, on a quick browse through that paper I did not see any mentioning of the Jensen wake model. The Jensen 1983 report is often cited together with the 1986 paper by Katic, Højstrup and Jensen, which introduces the method of wake superposition employed in the Park model in WAsP.

P2, L28: it is important to include more references on capturing multiple wakes in wind farms. At present only two are listed. But since this is the main focus of the paper it is crucial to establish the existing state of this area of study. "Among others" is insufficient. Here are some suggestions: ć Hirth et al., Wind Energy 18, 529 (2015) - Coupling Doppler radar-derived wind maps with operational turbine data to document wind farm complex flows ć Hirth et al., Wind Energy (2015) - Dual-Doppler measurements of a wind ramp event at an Oklahoma wind plant ć Kumer et al., Energy Procedia 80 245 (2015) - Characterisation of single wind turbine wakes with static and scanning WINTWEX-W LiDAR data (already cited elsewhere in the paper) ć Wang and Barthelmie paper mentioned above ć Van Dooren et al., Remote Sens. 2016, 8, 809 - A Methodology for the Reconstruction of 2D Horizontal Wind Fields of Wind Turbine Wakes Based on Dual-Doppler Lidar Measurements

P3, L12: a photo or photo collage would greatly assist the understanding of the setup of the field campaign, the description of the surroundings and possibly the interpretation of the results.

P3, Fig.1: the figure is good, but should only include the relevant information. Were all the surface flux stations used in determining the atmospheric stability? Were all profiling lidars used in the analysis? Leave out the details not connected with this paper.

Sec 2.1: the text should specify clearly how the profiling lidars were used in the analysis. Section 2.1 has a brief mention of a comparison between the scanning lidar and WC-3. How was this done? What did the results show? WC-3 was used to deter-

СЗ

mine veer on page 14, but please introduce this in the description of the observational dataset. Was WC-2 used for anything in the present study?

P4, L15: the period at the beginning of the line belongs at the end of the previous line Sec.2.1.1: I miss a detailed description of the scan patterns. What was the sector size for the PPI scnas? What was the time per scan? This is hinted at on P14, but it belongs in this section. What was the order of the RHI and PPI scnas? Am I right that the RHI scans are not used in the present analysis? If that is correct, then please state that. Was the pointing accuracy of the scanning lidar checked using hard target returns (eg from the wind turbines)? This information is important to interpret the results and should be included. Alternatively, if the authors have made this information available elsewhere, a reference could suffice.

P5, Fig. 2: how large was the change in elevation between the lidar location and the turbines? State what is was, then argue why it can be neglected. Otherwise, make an assessment of the uncertainty or bias it introduces into the results.

P6, L3: the Monin-Obukhov length has dimensions of Length. I assume the listed limits should be in meter.

P6, L6: insert a "The" at the beginning of the line. Same on line 23 after Figure 3.

P6, L12: define carrier-to-noise ratio. Explain the filter on CNR (<-27dB) and why this is necessary. Why was this threshold chosen? How sensitive are the results to this value?

P6, L13: do data refer to the line of sight wind speeds?

P6, L13: define \mu and define MAD in an equation. Is the standard deviation equal to MAD (it is cryptically stated to be evaluated according to MAD)? How sure are you that the outliers removed are not physical? Do you know the source of the outliers you exclude?

P6, L23: I suggest inserting "the assumed" in front of uniform ambient wind speed.

P7, L1: there is a "the" missing in front of ambient flow speed.

P7: it would be very useful to have an overview image or map showing the turbine locations and the scanned sector.

P7, Fig.3: to define the wind direction there should be an indication of the north (or south) direction in the figure. Is south upwards in the figure?

P9, L19: MAD acronym was already defined. When the method was applied again, did you use the same bounds as on page 6?

P9, L20: "mean characteristic" – is this for the entire database or for a single scan?

P9 L21: define the Pearson correlation coefficient and the mean squared error.

P10, Fig. 5: what do the corresponding plots of data availability look like? These could be important to include to understand if the some of the decrease in Figure 5 is driven by the measurements and not by wake characteristics. I would also like to know the number of scans included in the stable and unstable curve. Were there no neutral conditions?

P11, Fig. 6: are these plots along the wake centerlines? What was the wind direction and how was it oriented with respect to the turbine row? This can be deduced from figures 8 and 9, but you might as well make it clear here.

P11, Fig. 7: the authors should attempt to explain the differences they see. While the two outer turbines both have smaller deficits than the inner turbines, the difference in deficit between the two outer turbines is larger than the difference between the inner turbine and the outer turbine with the highest deficit. It would be useful to label the curves with the turbines they belong to. Could there be a relation with the vertical structure of the wakes and sampling the wakes at different heights. As stated above it is important to know how the different elevation tilts were combined (or not) in making this figure. Is the width of the shaded bands one or two standard deviations? I suggest making the same plot for the 23 August data.

P 11, L 4: replace low with small

P12, L 12: the passive voice makes it hard to understand the sentence. Consider rephrasing it. I think you mean widest, when you write largest.

P16, Fig. 11: I am not sure I agree with the conclusion of a larger angular difference for the outer turbines based on the linear fits. The linear fits are very poor. Indeed, the data could be seen as describing an oscillation, where the inner and outer turbines follow each other.

P20, L16: & amp should be &.

P21, L14: Spera, D should be Neustadter, H. E. and Spera, D. and the page number is 240 not 241.

---

## Referee Comment (RC3) · Anonymous Referee #3 · 10 May 2017

The manuscript entitled "Three-dimensional structure of wind turbine wakes as measured by scanning lidar" deals with the analysis of field measurements performed during stable atmospheric conditions on the velocity field downstream of a group of four wind turbines, captured by a scanning LiDAR.

The study is of great interest for the wind energy community since it can contribute to better quantify the wind turbine wake properties and so, to validate some wake models and numerical simulations.

The experimental set-up is well detailed, the method to detect the wake locations of multiple wakes on each snapshot is well described and the content is well structured. On the other hand, the discussion is rather poor: several results, which are not intuitively expected, are mentioned but not justified. For instance:

- The reason why the velocity deficit is smaller for outer wakes than for inner wakes. Is it possible to give an explanation without having information about the wind turbine operating conditions? The velocity deficit is also primarily related to the power coefficient of the wind turbine.

- The reason why the wake width is dependent on the relative position between the wake and the scanning lidar. Please elaborate an explanation.

- The correlation between the veer and the wake stretching angle is rather poor (figure 11), the data present a very high scatter, with no linear trend. It is hazardous to make some interpretation with this plot. Parallel to the previous remark on the dependence of wake widths to lidar beam orientation, could the measurement set-up and the scanning lidar limitations be responsible of this wake stretching, instead of the veer? Again, the conclusion that the inner and outer wakes behave differently with the veer effect must be justified.

Minor comments:

- Page 4, line 12: 30-min cycle: do you mean that, during 30 minutes, several PPI and RHI are collected? If yes, please indicate the duration to collect one PPI and one RHI. It will give an idea of the temporal resolution of the obtained velocity field.

- Give the range of azimuth angles that have been scanned during PPI and RHI. A table with all these information would be appropriate.

- Page 5, figure 5: the y-axis legend indicates "detected wakes [% scans]" but the maximum value is 1, and not 100.

---

## Author Comment (AC1) · 12 Jun 2017

The comment was uploaded in the form of a supplement:
http://www.atmos-meas-tech-discuss.net/amt-2017-86/amt-2017-86-AC1-supplement.pdf

---

## Author Comment (AC2) · 12 Jun 2017

The comment was uploaded in the form of a supplement:
http://www.atmos-meas-tech-discuss.net/amt-2017-86/amt-2017-86-AC2-
supplement.pdf

---

## Author Comment (AC3) · 12 Jun 2017

The comment was uploaded in the form of a supplement:
http://www.atmos-meas-tech-discuss.net/amt-2017-86/amt-2017-86-AC3-supplement.pdf

---

## Author Response (AR1)

**1. Review #1**

*In this document, the reviewer comments are in black, the authors responses are in red.*

The authors thank the reviewer for their thoughtful and productive comments.

SUMMARY: This paper discusses the analyses of wind turbine wake structure considering varying atmospheric stability regimes using measurements from a single scanning lidar; representing an extension of the work of Aiken and Lundquist (JTECH, 2014). Frequency in wake detection, wake velocity deficits, wake width, and wake centerline results are presented. I found the discussion related to the wake stretching as a result of vertically veering wind direction to particularly interesting. The impact of atmospheric stability on wake behavior is a very important aspect to understanding wind plant performance as a whole, and as this paper shows there are significant and consistent (e.g. diurnal) changes in wake behavior as a result of the background stability. I believe the results of this paper are a meaningful contribution and worthy of publication, but I do have a few comments/questions requesting clarity for how the measurements are used to construct the analyses and what the downstream implications are for the presented results.

Thank you for finding our work interesting and useful!

MAJOR COMMENTS/QUESTIONS: While the title infers a 3D construction of the wakes for analysis, in reality, all of the presented wake statistics are only assessed from individual 2D planes. Since multiple elevation tilts are used, I would have liked to see the author construct fully integrated 3D volumes of data by interpolating the polar data from a given elevation series (by elevation series I mean a single collection of the six elevation tilts between 1.5-2.8 degrees) into a 3D Cartesian grid. Otherwise, when assessing wake deficit and width, how can we be certain that any given 2D plane is a complete representation of the absolute downwind wake deficit or wake width since that plane only represents a horizontal 2D "slice" somewhere through the wake?

You are right: single 2D plans cannot be considered as a full representation of wake characteristics. In fact, we considered data from PPI scans performed at all the six elevation angles to retrieve results for the velocity deficit and the wake width. As shown in Figure 2, we can in this way produce a representation of a considerable part of the whole wake region, at different altitudes and different downwind distances from the turbine.

Since deficits and widths were computed separately from the individual PPI elevation planes within the same elevation series, this would mean multiple wake deficit/width calculations for the same downwind distance exist within the same elevation series, but inherently represent different vertical locations within the wake. Or maybe the generated wake statistics for one elevation series

are binned by range regardless of their height?

For the whole discussion of the velocity deficit and the wake width, all the different elevation angles of the PPI scans are included in the results: the focus for these two characteristics is mainly on how these change with downwind distance and how different relative position (i.e. inner vs outer wakes) affect the results.

Additional discussion on how the statistics from individual tilts within a single elevation series are merged (or not) would provide for a better understanding of the bulk wake statistics (e.g. deficit and width) and how representative the statistics are of the character of the full wake at any given downwind distance.

Thank you for noticing that our discussion about the different elevation angles was not clear. The revised manuscript will clearly state that all the six elevation angles are used to retrieve the results regarding wake width and velocity deficit: "during all the scans (at all the considered elevation angles) performed on 23 and 26 August 2013".

I apologize if I have misinterpreted the analysis, but I believe there needs to be further clarification of how the 2D planes of PPI data are used to generate the wake statistics, and what inherent assumptions/limitations are associated with the methods. I would also encourage the author, though not required, to consider analyzing the data in a true 3D framework when constructing the presented wake statistics.

Although we appreciate the suggestion of a full 3D framework retrieval of the wake structure, we think that the way we chose to present our results – now improved to better explain how we used different elevation angles – is more intuitive and similar to the presentation style of comparable results in the literature in this field (Aitken et al. 2014, Banta et al. 2015, among the others). Thus, we decided to keep our way to present the results; however, we thank the reviewer for their useful suggestion, which might be implemented in a future paper about the topic, possibly using data from different field campaigns.

MINOR COMMENTS/QUESTIONS:

1. P4, L8: What is the range from the WINDCUBE to the four turbines of interest? The distances can be inferred from Figure 2, but the numbers would be useful in the text.

The revised caption of Figure 2 will include the distances of the four considered turbines from the scanning lidar (2136 m, 2102 m, 2171 m, and 2286 m).

2. P 4, L 14: What is the scan speed of the lidar? What range of azimuths are scanned?

The scan speed of the lidar during a PPI scans was of 0.5deg/s. Each PPI scan spanned an azimuth range of 50deg. The revised manuscript will include the following sentence: "each PPI scan lasted approximately 100 seconds, spanning an azimuth range of 50 deg with a speed of 0.5 deg/s."

3. P4, L 19: How long does it take the lidar to scan the series of six elevation tilts?

The following table with a detailed description of the scan pattern – and its duration - used in the experiment will be included in the manuscript, along with an explicit statement that "Approximately ten minutes were required to collect the series of six elevation tilts":

**Table 2.** *Description of the 30-min cycle of scanning lidar scans in CWEX-13 field campaign. The characteristic fixed angle refers to the elevation angle for PPI and VAD scans, the azimuth angle for RHI scans.*

| number of scans | type of scan | characteristic fixed angle | duration of each scan | cumulative time |
|---|---|---|---|---|
| 2 | VAD | $75°, 60°$ | 132 s | 0:00 - 4:24 |
| 6 | PPI | $2.8°, 2.5°, 2.2°, 2.1°, 1.8°, 1.5°$ | 104 s | 4:24 - 14:48 |
| 3 | RHI | $160°, 170°, 180°$ | 32 s | 14:48 - 16:24 |
| 6 | PPI | $2.8°, 2.5°, 2.2°, 2.1°, 1.8°, 1.5°$ | 104 s | 16:24 - 26:48 |
| 6 | RHI | $160°, 170°, 180°, 180°, 170°, 160°$ | 32 s | 26:48 - 30:00 |

4. P9, L 11: This comment relates to the Major Comments/Questions section above. How is the ambient flow wind speed defined on the 2D PPI plane? Since the PPI plane is slanted, if the ambient flow wind speed is determined upstream of a given turbine, can you comment on the impact of using this value for constructing wake deficits downstream of the turbine, but at lower heights due to the slanted PPI plane? Is the comparison being made at different heights because the PPI plane is sloped? If so, what are the implications on the wake deficit calculations?

Thank you for pointing out that we did not specify which ambient flow speed we used for this calculation. The ambient flow wind speed is estimated (as one of the parameters of the used models) by our wake characterization algorithm at each range gate in each PPI scan. Thus, since we have a different value for each range gate, different vertical levels do not affect this calculation. To make this clear, we will rephrase the sentence as follows: "the ambient flow wind speed u (estimated from our algorithm at each performed fit at each range gate and elevation)".

5. P9, L30: How many scans are used to generate the statistics in Figure 5? Are all of the 438/576 scans performed on 23 Aug/26 Aug considered? Are all elevation tilts considered?

Yes, all the scans at all elevation angles are considered in this Figure. Thanks for pointing out that this had not been clearly stated in the text. The following sentence will be added to clarify this: "during all the 438/576  scans (at all the considered elevation angles) performed on 23/26 August

2013".

6. P10, L13: Does this statement imply that a single elevation series of six scans takes 11 minutes to complete? If so, that may answer Question 3 above.

Yes it does. However, as stated above, the duration of the scans will be explicitly shown in a table in the revised manuscript.

7. P10, L18: While the difference in wake deficit between the inner and outer wakes is shown, can the author comment on the difference in wake deficit between the outer turbines, as that difference is actually more substantial? Which line is for which turbine? Assuming the wind direction is relatively consistent throughout (maybe a bad assumption), can a composite PPI image be provided corresponding with this period to highlight if there are any features in the flow (e.g. a turbine row edge effect) that could be contributing?

See our answer to the next comment below.

8. P12, L2: While the author states comparable results occur between the 23-August and 26-August cases, do the red lines, for example, flip with the different wind direction, again inferring some type of turbine row edge effect?

Thank you for your thoughts about this result. We will change our Figure to label the individual turbines, as follows:

[Figure]

The sketch on the right will also help the reader to remember the wind direction for the considered

day.

Thank you also for pointing out that our comparison between 23 and 26 August was not clear enough. To solve this issue, we will include the plot for 23 August (see below) in Figure 7, whose caption will be rephrased as: "Velocity deficit vs downwind distance, for the four wakes of the studied row of turbines. Continuous lines represent the median values calculated from the PPI scans performed at all the considered elevation angles during the night (stable conditions) of 26 (panel a) and 23 (panel b) August 2013".

[Figure]

As can be seen, for the 23 August 2013 the difference between outer (turbines 1 and 4) and inner (turbines 2 and 3) wakes is more consistent for all the studied turbines, and the difference between the velocity deficit for the two outer turbines (noticed by the reviewer for the 26 August case) is not present anymore. Therefore, we cannot infer general results for other flow features (e.g. edge effects) beyond what we stated in the paragraph: "wakes from outer turbines (number 1 and number 4), have lower velocity deficits than the wakes from inner turbines (number 2 and number 3), for relatively small downwind distances, with a difference up to 15%".

9. P12, L18: Could an example PPI image or composite be included to visually highlight the wind direction dependence on wake width detection being discussed here?

This dependence can be seen in the following two color-maps of the line-of-sight velocity measured in two PPI scans during 23 and 26 August, which will be included in the manuscript at the end of the "Lidar measurements" section:

[Figure]

The caption of the Figure will be: "Figure 3. Color maps of line-of-sight velocity measured by the scanning lidar during two PPI scans performed at 11:57 UTC (6:57 am LDT) on 23 August 2013 (panel a) and at 02:33 UTC (9:33 pm LDT) on 26 August 2013 (panel b). The scanning lidar is located in the origin of the coordinate system. The two arrows show wind direction as measured by the profiling lidars WC-1 and WC-2 at 80 m AGL."

Moreover, we will add the following paragraph and Figure to better explain our interpretation of this result:

"This result is due to the relationship between the viewing angle and the aspect ratio of the lidar retrieval "pixels", which are related to the relatively long range gate (50 m) and relatively narrow azimuthal resolution (0.5 degree). As qualitatively shown in the schematic of Figure 10, the scanning lidar measures the line-of-sight velocity in narrow pencil-shaped "pixels". With this geometry, if the wind direction - and thus the wake - is aligned with the line-of-sight from the lidar, the wake width can be assessed with high precision due to the high azimuthal resolution in each pencil-shaped area (panel a). However, if the wind direction - and thus the wake - is not aligned with the line-of-sight from the lidar (panel b), then the same wake will be measured as generally wider, since the retrieval of the wake width is now affected by the relatively coarse radial resolution of the lidar coordinate grid. In the schematic diagram shown in Figure 10, at an arbitrary fixed downwind distance from the turbine, the (same) wake would be detected as 19% larger when it is not aligned with the line-of-sight from the scanning lidar. This result becomes more evident the when the laser beam is more perpendicular to the wake. This result is due to the aspect ratio of the lidar "pixels," and thus would affect other wake characterization approaches relying on instruments not co-located with the turbine such as in Banta et al. (2015); Aitken et al. (2014a), but would not affect nacelle-mounted wake measurements, such as in Bingöl et al. (2010); Aitken and Lundquist (2014) as nacelle-mounted wake measurements are usually aligned with the wake,

unless the wake is intentionally yawed (Fleming et al., 2016; Trujillo et al., 2016)."

[Figure]

"Figure 10. Qualitative sketch of the dependence of detected wake width on the orientation of the coordinate grid used by a scanning lidar (purple triangle) as a function of the wind direction. Panel a shows the case of a wake aligned with the line-of-sight from the scanning lidar (wind direction shown by the blue arrow), while panel b shows the case of a wake not aligned with the line-of-sight from the lidar. The dashed arrow highlights the difference in the detected wake width for the two cases, at fixed downwind distance from the turbine (yellow ellipse)."

10. P12, Section 4.4: This comment relates to the Major Comments/Questions section above. Perhaps consider constructing a 3D volume of interpolated information, as opposed to compositing two horizontal planes for comparison. The change in wake centerline with height, supported by the presented measurements, is a really neat result of this study. The actual shape of the ellipsoid could potentially be better described (and compared to previous measurements) if the data were constructed in a 3D framework.

As already mentioned before, we thank the reviewer for their suggestion of a 3D volume of interpolated data; however, we decided not to implement this in our present work. At any rate, the results presented in Section 4.4 do NOT compare two horizontal planes, but two vertical regions, as shown in Figure 9 and in Figure 11, which we think is much more simple and intuitive than a real 3D framework to show our main finding of the vertical stretching of the structure of a wake, but with a lower magnitude than the ambient veer.

11. P12, L28: Why were the data separated between 55-75 m and >75 m, especially given hub height is at 80 m and data exist below 55 m within the rotor sweep. Why was the lower region bound vertically by 20 m but the upper region allowed to be larger? A quick comment on why this method was chosen could be beneficial.

The lower bin of data is between 35 and 55 m, and not between 55 and 75 as stated here. With this choice, we can have a great number of positions at low and high levels compared with the vertical dimensions of the turbines. By choosing these levels, we have an approximately equal number of vertical positions in each bin, given the geometry shown in Figure 2. To make this clear, we will include the following sentence: "these levels were chosen to create bins at low and high heights, compared to the vertical dimension of the turbines, with approximately the same number of vertical positions where the lidar measurements were taken, as shown in Figure 2".

MINOR EDITS:

1. P2, L16: "four" instead of 4.

2. P4, L15: The period at the start of this line should be at the end of the previous line.

3. P4, L23: "six" instead of 6.

4. P6, L14: Insert a period after (MAD).

5. P7, L11: Insert the word "the" between "as independent".

6. P9, L19: MAD is already defined on P6.

Thank you for catching all these edits! The manuscript will be changed accordingly.

**2. Review #2**

*In this document, the reviewer comments are in black, the authors responses are in red.*

The authors thank Dr. Nygaard for his detailed review and useful suggestions to improve the quality of our work.

Summary: this paper presents single lidar measurements of multiple wind turbine wakes. A wind field reconstruction model is used to derive wake characteristics from the line of sight wind speeds. The presented model is an extension to multiple wakes of a previous model developed by the same research group to characterize a single wake. Results are presented for the decay of the velocity deficit and the increase of the wake width with downstream distance. The wake centerline is found to shift orientation with height, indicating stretching of the wake profile in veered flow. The results are interesting and add to the developing picture of wakes in wind farms.

Thank you for finding our results interesting!

The method is sound, but could have been better explained, especially with regards to the vertical dimension and the influence of multiple elevation tilts (see specific comments below). The authors find some differences between the outer and the inner turbines in a row, but no explanation for these differences is offered.

The paper is definitely worthy of publication, but I suggest the authors consider the questions and suggestions below to improve the presentation of the methodology and the results.

Questions and suggestions:

In the model of Eq. (4) there is no reference to the vertical dimension or the elevation angle. Is it supposed to be applied at a fixed elevation angle? It is unclear how the vertical structure of the wakes is considered. At a fixed elevation angle the laser beam will probe at increasing height with increasing range. Were multiple elevation tilts combined in figures 7 and 8 to account for this?

Yes, equations (3) and (4) are applied at each fixed elevation angle. Then, to compute results in Figure 7 and 8, all the six elevation angles used in the campaign are combined to produce the final results.

To make this clear, we will modify the sentence that introduces our models as follows: "At a fixed elevation angle, the line-of-sight velocity u_LOS can be related to …".

Moreover, to make clear that Figures 7 and 8 use data from all the elevation angles, we rephrased the sentences below as follows:

- "Figure 7 shows velocity deficit versus downwind distance from the turbines, calculated from the 276 PPI scans (at all the elevation angles) performed during the whole night (stable conditions) of 26 August 2013."
- [Caption of Figure 7] "Continuous lines represent the median values calculated from the PPI scans performed at all the considered elevation angles during the night (stable conditions) of 26 (panel a) and 23 (panel b) August 2013".
- [Caption of Figure 8] "Wake width vs downwind distance from the turbines, for the wakes of the four turbines in the studied row, from PPI scans performed at all the six considered elevation angles."

Consider adding a reference to Wang and Barthelmie - Journal of Physics: Conference Series 625 (2015) 012017 - Wind turbine wake detection with a single Doppler wind lidar. This has a similar wake wind field reconstruction method.

We will add a reference to this paper in the introduction.

P1, L23: The reduction in power for turbines in wake can exceed the 40% mentioned as the upper limit. As is well known, it is very sensitive to wind direction, being largest when the wind is aligned with rows of turbines. Also the observed maximum reduction depends on the size of the wind direction sector over which the data are averaged. But even for a 30 degree sector Nygaard 2014 found reductions of up to 60% for certain conditions. The total wake loss considering all wind directions and wind speeds is typically less than 20%. My point is that the 40% mentioned in the text is a meaningless number without further context. It only applies for certain wind directions and for averaging over a sector of a certain size. I invite you to make the context of this number clearer or to consider, if a specific value is needed.

Thank you for pointing out that we should have been more careful with this. Since, as you explained, giving a precise number for this energy reduction is not possible, we will eliminate this sentence from the manuscript, and include the references to Barthelmie et al. 2010 and Nygaard 2014 in the previous sentence, along with other papers which deal with energy reductions in wakes.

P2, L11: the appropriate reference for the Jensen model is N. O. Jensen, A note on wind generator interaction, Risø-M-2411 (1983). The reference you have to Jensen 1984 was new to me, so thank you for pointing it out. However, on a quick browse through that paper I did not see any mentioning of the Jensen wake model. The Jensen 1983 report is often cited together with the 1986 paper by Katic, Højstrup and Jensen, which introduces the method of wake superposition employed in the Park model in WAsP.

Thank you for the useful suggestion. We will modify the reference, which will include Jensen

1983 and Katic et al. 1986.

P2, L28: it is important to include more references on capturing multiple wakes in wind farms. At present only two are listed. But since this is the main focus of the paper it is crucial to establish the existing state of this area of study. "Among others" is insufficient. Here are some suggestions: Hirth et al., Wind Energy 18, 529 (2015) - Coupling Doppler radar-derived wind maps with operational turbine data to document wind farm complex flows, Hirth et al., Wind Energy (2015) - Dual-Doppler measurements of a wind ramp event at an Oklahoma wind plant, Kumer et al., Energy Procedia 80 245 (2015 ) - Characterisation of single wind turbine wakes with static and scanning WINTWEX-W LiDAR data (already cited elsewhere in the paper), Wang and Barthelmie paper mentioned above, Van Dooren et al., Remote Sens. 2016, 8, 809 - A Methodology for the Reconstruction of 2D Horizontal Wind Fields of Wind Turbine Wakes Based on Dual-Doppler Lidar Measurements

Thank you for pointing out that we should improve the references here. We will substantially improve this paragraph with the references you suggest: "The interactions between multiple wakes must be captured in studies of large wind farms, as done by Clive et al. (2011); Hirth et al. (2015a, b); Kumer et al. (2015); Wang and Barthelmie (2015); Aubrun et al. (2016); van Dooren et al. (2016).".

P3, L12: a photo or photo collage would greatly assist the understanding of the setup of the field campaign, the description of the surroundings and possibly the interpretation of the results.

The following pictures showing the instruments at the site of the campaign (and its land use) will be included in the Supplementary Material.

[Figure]

*Figure 1: scanning lidar co-located with WC-3, looking to the southwest (photo courtesy Paul Quelet)*

[Figure]

*Figure 2: WC-1, looking towards SE (photo courtesy Lundquist)*

[Figure]

*Figure 3: WC-3 looking to SE (photo courtesy Lundquist)*

[Figure]

*Figure 4: WC-2 looking to SE (photo courtesy Paul Quelet)*

P3, Fig.1: the figure is good, but should only include the relevant information. Were all the surface flux stations used in determining the atmospheric stability? Were all profiling lidars used in the analysis? Leave out the details not connected with this paper.

Although we agree that the figure includes some instruments that were not directly used in our work, we still think that it is important to give a complete overview of the set-up of the instruments at the campaign. However, we will improve our description in the manuscript to make clear which instruments we used for our research.

Regarding the different surface flux stations, we will include the following sentence in Section 2.1.2 to clearly states which station we used in our work: "We used measurements from the surface flux station ISU_3 to assess atmospheric stability conditions, with the calculation of Obukhov length".

Regarding the different vertical profiling lidars, see the comment below.

Sec 2.1: the text should specify clearly how the profiling lidars were used in the analysis. Section 2.1 has a brief mention of a comparison between the scanning lidar and WC-3. How was this done? What did the results show? WC-3 was used to determine veer on page 14, but please introduce this in the description of the observational dataset. Was WC-2 used for anything in the present study?

Thank you for noticing that we should be more clear regarding how we used data from the different instruments deployed at the site. Regarding the comparison between WC-3 data and scanning lidar data, the manuscript already includes the reference to the Vanderwende's et al. paper where this comparison is described in detail. Our paragraph includes the following sentence about the results: "Vanderwende et al. (2015) demonstrated good agreement between the co-located scanning and WC-3 profiling lidar measurements at the altitudes where measurements overlapped."

Regarding how we used data from the different profiling lidar, we will add this sentence in the description of the observational dataset: "At CWEX-13, southerly wind conditions dominated the campaign. So, we used data from the WC-1 profiling lidar to measure upwind conditions for the studied row of turbines, and calculate the ambient wind veer." We did not use data from WC-2 and WC-3 in the retrievals of wake characteristics.

P4, L15: the period at the beginning of the line belongs at the end of the previous line Sec.2.1.1: I miss a detailed description of the scan patterns. What was the sector size for the PPI scans? What was the time per scan? This is hinted at on P14, but it belongs in this section. What was the order of the RHI and PPI scans? Am I right that the RHI scans are not used in the present analysis? If that is correct, then please state that. Was the pointing accuracy of the scanning lidar checked using hard target returns (eg from the wind turbines)? This information is important to interpret the results and should be included. Alternatively, if the authors have made this information available elsewhere, a reference could suffice.

Thank you for pointing out that we did not provide this piece of information in the proper location and that we should provide a more detailed description of the scan patterns. We will add the following sentence: "each PPI scan lasted approximately 100 seconds, spanning an azimuth range of 50deg with a speed of 0.5deg/s, while a RHI had a duration of about 30 seconds." We will also add the following table with more details about the scans performed during the field campaign:

**Table 2.** *Description of the 30-min cycle of scanning lidar scans in CWEX-13 field campaign. The characteristic fixed angle refers to the elevation angle for PPI and VAD scans, the azimuth angle for RHI scans.*

| number of scans | type of scan | characteristic fixed angle | duration of each scan | cumulative time |
|---|---|---|---|---|
| 2 | VAD | $75°, 60°$ | 132 s | 0:00 - 4:24 |
| 6 | PPI | $2.8°, 2.5°, 2.2°, 2.1°, 1.8°, 1.5°$ | 104 s | 4:24 - 14:48 |
| 3 | RHI | $160°, 170°, 180°$ | 32 s | 14:48 - 16:24 |
| 6 | PPI | $2.8°, 2.5°, 2.2°, 2.1°, 1.8°, 1.5°$ | 104 s | 16:24 - 26:48 |
| 6 | RHI | $160°, 170°, 180°, 180°, 170°, 160°$ | 32 s | 26:48 - 30:00 |

We did only use PPI scans to detect wakes, and this is stated at the beginning of section 3: "The line-of-sight velocity measured by the WINCDUBE 200S scanning lidar (Figure 3) during the horizontal (PPI) scans can be analyzed to determine wake characteristics and how they evolve in space as the wakes propagate."

P5, Fig. 2: how large was the change in elevation between the lidar location and the turbines? State what is was, then argue why it can be neglected. Otherwise, make an assessment of the uncertainty or bias it introduces into the results.

The change in elevation between the turbines and the scanning lidar was 7m. This will be included in the caption of the figure. However, we DID take into account this difference, as already stated in the caption itself.

P6, L3: the Monin-Obukhov length has dimensions of Length. I assume the listed limits should be in meter.

Thank you for noticing this. We will correct the manuscript accordingly.

P6, L6: insert a "The" at the beginning of the line. Same on line 23 after Figure 3.

We will add "the" in both the proposed sentences.

P6, L12: define carrier-to-noise ratio. Explain the filter on CNR (<-27dB) and why this is necessary. Why was this threshold chosen? How sensitive are the results to this value?

We will add more details regarding the CNR threshold: "First, a threshold is imposed to the carrier-to-noise ratio (CNR), which represents the strength of the backscattered signal compared to background noise (values closer to 0 dB indicate a stronger signal relative to the noise): all measurements with carrier-to-noise ratio CNR < -27 dB are discarded from further analysis [Vanderwende et al. 2015]. Measurements with a lower CNR often had unrealistically high (> 15

m/s) values of radial velocity; and this threshold value is comparable with choices in other studies [Cariou et al. 2011, Bastine et al. 2015, Debnath et al. 2016]."

P6, L13: do data refer to the line of sight wind speeds?

Yes. To make this clear, we will change the sentence in "…line-of-sight velocity data…".

P6, L13: define \mu and define MAD in an equation. Is the standard deviation equal to MAD (it is cryptically stated to be evaluated according to MAD)? How sure are you that the outliers removed are not physical? Do you know the source of the outliers you exclude?

We will improve the description of the MAD method as follows: "in each scan, line-of-sight velocity data which are not included in the interval $(\mu - 3\hat{\sigma}, \mu + 3\hat{\sigma})$, where $\mu$ is the average of the data, are removed from the analysis. The standard deviation $\hat{\sigma}$ is evaluated according to the median absolute deviation (MAD), assuming normally distributed data: $\hat{\sigma} = 1.4826$ MAD, where $MAD = \text{median}(|u_{LOS,i} - \text{median}(u_{LOS})|)$".

The threshold was chosen as in Aitken et al. 2014, and, under the assumption of normally distributed data, 99.7% of data are included in the selected interval. Thus, the discarded values can be considered to be due to extreme events such as hard strikes.

P6, L23: I suggest inserting "the assumed" in front of uniform ambient wind speed.

Thank you for the suggestion. We will add "the assumed" in the sentence.

P7, L1: there is a "the" missing in front of ambient flow speed.

We will add "the" in the sentence.

P7: it would be very useful to have an overview image or map showing the turbine locations and the scanned sector.

The following figure, showing the location of the scanning lidar, of the four turbines, and the line-of-sight velocity measurements during two PPI scans (one for each of the studied days), will be added to the manuscript at the end of the "Lidar measurements" section. The Figure will be introduced in the paragraph as follows: "Figure 3 shows examples of maps of line-of-sight velocity measured by the scanning lidar during two PPI scans performed at night on the selected days. The wind turbine wakes can clearly be detected in terms of reduced wind speed downwind of the four wind turbines."

[Figure]

The caption of the Figure will be: "Figure 3. Color maps of line-of-sight velocity measured by the scanning lidar during two PPI scans performed at 11:57 UTC (6:57 am LDT) on 23 August 2013 (panel a) and at 02:33 UTC (9:33 pm LDT) on 26 August 2013 (panel b). The scanning lidar is located in the origin of the coordinate system. The two arrows show wind direction as measured by the profiling lidars WC-1 and WC-2 at 80 m AGL."

P7, Fig.3: to define the wind direction there should be an indication of the north (or south) direction in the figure. Is south upwards in the figure?

North is upwards in the Figure. To make this clear, we will modify the Figure adding a North direction arrow, and rephrase a sentence as follows: "…both alpha and phi are >0 for clockwise rotations from North."

P9, L19: MAD acronym was already defined. When the method was applied again, did you use the same bounds as on page 6?

We will eliminate "median absolute deviation" from the sentence since we already defined MAD previously. Yes, we used again 3 standard deviations as threshold. The sentence will be rephrased as: "the MAD method is applied again to discard wake characteristics which do not lie within three standard deviations of the mean".

P9, L20: "mean characteristic" – is this for the entire database or for a single scan?

Thank you for noticing that we didn't specify this clearly. The sentence will be rephrased as: "the mean characteristic at each range gate for each whole night".

P9 L21: define the Pearson correlation coefficient and the mean squared error.

We will rephrase the sentence as follows:

with Pearson correlation coefficient ($corr(u_{LOS}, \hat{u}_{LOS}; g) = cov(u_{LOS}, \hat{u}_{LOS}; g)/\sqrt{cov(u_{LOS}, u_{LOS}; g)cov(\hat{u}_{LOS}, \hat{u}_{LOS}; g)}$, where $g$ represents the data weights) larger than 0.9 and mean squared error ($MSE = \frac{1}{\sum_{i=1}^{n} g_i} \sum_{i=1}^{n} g_i(\hat{u}_{LOS,i} - u_{LOS,i})^2$) lower than 0.5 are included in the final analysis of the results.

P10, Fig. 5: what do the corresponding plots of data availability look like? These could be important to include to understand if the some of the decrease in Figure 5 is driven by the measurements and not by wake characteristics. I would also like to know the number of scans included in the stable and unstable curve. Were there no neutral conditions?

Since we considered southwesterly (26 August 2013) and southeasterly (23 August 2013) flows, as the wakes go further from the turbines, they actually get closer to the scanning lidar (see scheme in Figure 1), so we are not concerned about data availability at long downwind distances (i.e. close to the lidar). However, the following plot showing data (line-of-sight velocity) availability for 23 and 26 August will be included in the Supplementary Material. The vertical dashed lines show the positions of the turbines. As can be seen, the data availability for all the range gates considered in Figure 5 (i.e. to the left of the dashed lines) is nearly 100%, and so our results are not affected by this possible issue.

[Figure]

Regarding neutral conditions, we will add the following sentence to the caption of the Figure: "(neutral conditions were detected only for very short periods, and are not included here)". So, nearly all the PPI scans performed during the selected days are included in Fig. 5.

P11, Fig. 6: are these plots along the wake centerlines? What was the wind direction and how was

it oriented with respect to the turbine row? This can be deduced from figures 8 and 9, but you might as well make it clear here.

Considering how we modeled each wake (equation 4) and how we defined the velocity deficit (equation 5) in the wake, the velocity deficit is by definition calculated at the location of the center of the wake – thus the wake centerline. Regarding the wind direction, it is stated in Section 2.1.1 that 26 August shows predominant southwesterly wind. However, as a reminder, the caption of the figure will include: "… 26 August 2013, southwesterly wind, …".

P11, Fig. 7: the authors should attempt to explain the differences they see. While the two outer turbines both have smaller deficits than the inner turbines, the difference in deficit between the two outer turbines is larger than the difference between the inner turbine and the outer turbine with the highest deficit. It would be useful to label the curves with the turbines they belong to. Could there be a relation with the vertical structure of the wakes and sampling the wakes at different heights. As stated above it is important to know how the different elevation tilts were combined (or not) in making this figure. Is the width of the shaded bands one or two standard deviations? I suggest making the same plot for the 23 August data.

We will change the Figure as follows to highlight the behavior of the single turbines:

[Figure]

And we will include the following plot from 23 August:

[Figure]

The paragraph which introduces this new version of the Figure will be changed accordingly: "Figure 7 shows velocity deficit versus downwind distance from the turbines, calculated from the 276 (242) PPI scans, at all the elevation angles, performed during the whole night - stable conditions - of 26 (23) August 2013."

As can be seen, for the 23 August 2013 the difference between outer (turbines 1 and 4) and inner (turbines 2 and 3) wakes is more consistent for all the studied turbines, and the phenomenon noticed by the reviewer does not appear. Therefore, we cannot infer general results beyond what we stated in the paragraph: "wakes from outer turbines (number 1 and number 4), have lower velocity deficits than the wakes from inner turbines (number 2 and number 3), for relatively small downwind distances, with a difference up to 15%".

Regarding the use of different elevation angles, the caption of the Figure will include: "Velocity deficit vs downwind distance, for the four wakes of the studied row of turbines. Continuous lines represent the median values calculated from the PPI scans performed at all the considered elevation angles during the night (stable conditions) of 26 (panel a) and 23 (panel b) August 2013;".

Regarding the shaded bands, we will rephrase the description in the caption as "shaded areas show $\pm$ one standard deviation of the data" (and the same was added for the caption of Figure 8).

P 11, L 4: replace low with small

We will replace the adjective. Thank you for suggesting!

P12, L 12: the passive voice makes it hard to understand the sentence. Consider rephrasing it. I think you mean widest, when you write largest.

We are not sure which passive voice you refer to. In any case, we will slightly rephrase the sentence as: "…the scanning lidar systematically identifies as the widest the wake which, at a given downwind distance, is the most perpendicular to the laser beam…".

P16, Fig. 11: I am not sure I agree with the conclusion of a larger angular difference for the outer turbines based on the linear fits. The linear fits are very poor. Indeed, the data could be seen as describing an oscillation, where the inner and outer turbines follow each other.

We agree that the quality of the linear fits is rather poor, however we think they are useful to show the different behavior between wakes from inner and outer turbines. To make this clear and to include your suggestion about oscillations, we will rephrase the description of these results as follows: "as suggested by the linear regression fits, wakes from outer turbines often present a larger angular difference in wake centerlines compared to wakes from inner turbines, though with variability for different veer values that motivates further study".

P20, L16: & should be &.

Thank you for catching this mistake!

P21, L14: Spera, D should be Neustadter, H. E. and Spera, D. and the page number is 240 not 241.

Thank you for noticing. We will correct the reference accordingly.

**3. Review #3**

*In this document, the reviewer comments are in black, the authors responses are in red.*

The manuscript entitled "Three-dimensional structure of wind turbine wakes as measured by scanning lidar" deals with the analysis of field measurements performed during stable atmospheric conditions on the velocity field downstream of a group of four wind turbines, captured by a scanning LiDAR.

The study is of great interest for the wind energy community since it can contribute to better quantify the wind turbine wake properties and so, to validate some wake models and numerical simulations.

The authors thank the reviewer for their time in reviewing this contribution and we are pleased that this work is considered of great interest.

The experimental set-up is well detailed, the method to detect the wake locations of multiple wakes on each snapshot is well described and the content is well structured. On the other hand, the discussion is rather poor: several results, which are not intuitively expected, are mentioned but not justified. For instance:

Thank you for pointing out that the discussion of the results should be improved. We will justify our results as explained in the next paragraphs.

- The reason why the velocity deficit is smaller for outer wakes than for inner wakes. Is it possible to give an explanation without having information about the wind turbine operating conditions? The velocity deficit is also primarily related to the power coefficient of the wind turbine.

To provide a possible explanation of this result, we will include the following sentence at the end of Section 4.2: "The presence of outer turbines seems to reduce the effectiveness of lateral entrainment of faster air to recover wind conditions in the inner wake regions of the wind farm." Moreover, to provide additional insight, the plot with the results for the 23 August will be included in the revised manuscript.

The turbines are all the same model of turbine and therefore would have the same power coefficient at each wind speed regime, and we have verified (not shown as the data are not publicly available) that the turbines are all producing, on a daily average basis, within 15kW and 31kW of each other during 23 and 26 August 2013, respectively.

- The reason why the wake width is dependent on the relative position between the wake and the

scanning lidar. Please elaborate an explanation.

Thank you for pointing out that we did not provide a sufficient discussion for this. We will add the following paragraph and Figure to our manuscript:

"This result is due to the relationship between the viewing angle and the aspect ratio of the lidar retrieval "pixels", which are related to the relatively long range gate (50 m) and relatively narrow azimuthal resolution (0.5 degree). As qualitatively shown in the schematic of Figure 10, the scanning lidar measures the line-of-sight velocity in narrow pencil-shaped "pixels". With this geometry, if the wind direction - and thus the wake - is aligned with the line-of-sight from the lidar, the wake width can be assessed with high precision due to the high azimuthal resolution in each pencil-shaped area (panel a). However, if the wind direction - and thus the wake - is not aligned with the line-of-sight from the lidar (panel b), then the same wake will be measured as generally wider, since the retrieval of the wake width is now affected by the relatively coarse radial resolution of the lidar coordinate grid. In the schematic diagram shown in Figure 10, at an arbitrary fixed downwind distance from the turbine, the (same) wake would be detected as 19% larger when it is not aligned with the line-of-sight from the scanning lidar. This result becomes more evident the when the laser beam is more perpendicular to the wake. This result is due to the aspect ratio of the lidar "pixels," and thus would affect other wake characterization approaches relying on instruments not co-located with the turbine such as in Banta et al. (2015); Aitken et al. (2014a), but would not affect nacelle-mounted wake measurements, such as in Bingöl et al. (2010); Aitken and Lundquist (2014) as nacelle-mounted wake measurements are usually aligned with the wake, unless the wake is intentionally yawed (Fleming et al., 2016; Trujillo et al., 2016)."

[Figure]

"Figure 10. Qualitative sketch of the dependence of detected wake width on the orientation of the coordinate grid used by a scanning lidar (purple triangle) as a function of the wind direction. Panel (a) shows the case of a wake aligned with the line-of-sight from the scanning lidar (wind direction

shown by the blue arrow), while panel (b) shows the case of a wake not aligned with the line-of-sight from the lidar. The dashed arrow highlights the difference in the detected wake width for the two cases, at fixed downwind distance from the turbine (yellow ellipse)."

- The correlation between the veer and the wake stretching angle is rather poor (figure 11), the data present a very high scatter, with no linear trend. It is hazardous to make some interpretation with this plot. Parallel to the previous remark on the dependence of wake widths to lidar beam orientation, could the measurement set-up and the scanning lidar limitations be responsible of this wake stretching, instead of the veer? Again, the conclusion that the inner and outer wakes behave differently with the veer effect must be justified.

We agree that the quality of the linear fits is rather poor, however we think they are useful to show the different behavior between wakes from inner and outer turbines. To make this clear and to include the suggestion of another reviewer about oscillations in this relationship, we will rephrase the description of these results as follows: "as suggested by the linear regression fits, wakes from outer turbines often present a larger angular difference in wake centerlines compared to wakes from inner turbines, though with variability for different veer values that motivates further study".

We do not think that the measurement set-up may have affected this result, since the same results are obtained for both 23 and 26 August (i.e. for different wind directions), while for the wake width different wind directions caused different results.

To provide a possible explanation for this result, we will add the following paragraph at the end of Section 4.4.1: "A possible physical explanation for this phenomenon can be detected in the interaction between wake rotation due to rotating blades and wind veer. The blades of the wind turbines in CWEX-13 wind farm rotate clockwise and so the downwind wakes rotates counter-clockwise [Burton 2001]. The wake can thus be considered as a sort of plume with its own momentum and rotation, which interacts with the ambient veer, that in turns tends to rotate the wake in the opposite direction (in the Northern hemisphere), thus causing a reduction in the global wake vertical stretching than would be present if only the ambient veer affected the wake. Inner wakes seem to be less subject to the effect of ambient wind veer, as if the presence of outer turbines reduces the ability of ambient wind characteristics to reach and impact inner regions of the wind farms."

Minor comments:

- Page 4, line 12: 30-min cycle: do you mean that, during 30 minutes, several PPI and RHI are collected? If yes, please indicate the duration to collect one PPI and one RHI. It will give an idea of the temporal resolution of the obtained velocity field.

Yes, several PPI and RHI scans are collected in each 30-min cycle. To make this clear, the revised text will include the following sentence: "each PPI scan lasted approximately 100 seconds, spanning an azimuth range of 50deg with a speed of 0.5deg/s, while a RHI had a duration of about 30 seconds". A table will also be included (see comment below) with more details about the 30-min cycle.

- Give the range of azimuth angles that have been scanned during PPI and RHI. A table with all these information would be appropriate.

The following table with a detailed description of the scans performed during the experiment will be included in the manuscript:

**Table 2.** *Description of the 30-min cycle of scanning lidar scans in CWEX-13 field campaign. The characteristic fixed angle refers to the elevation angle for PPI and VAD scans, the azimuth angle for RHI scans.*

| number of scans | type of scan | characteristic fixed angle | duration of each scan | cumulative time |
|---|---|---|---|---|
| 2 | VAD | $75°, 60°$ | 132 s | 0:00 - 4:24 |
| 6 | PPI | $2.8°, 2.5°, 2.2°, 2.1°, 1.8°, 1.5°$ | 104 s | 4:24 - 14:48 |
| 3 | RHI | $160°, 170°, 180°$ | 32 s | 14:48 - 16:24 |
| 6 | PPI | $2.8°, 2.5°, 2.2°, 2.1°, 1.8°, 1.5°$ | 104 s | 16:24 - 26:48 |
| 6 | RHI | $160°, 170°, 180°, 180°, 170°, 160°$ | 32 s | 26:48 - 30:00 |

- Page 5, figure 5: the y-axis legend indicates "detected wakes [% scans]" but the maximum value is 1, and not 100.

Thanks for noticing this incongruity. We will revise the plot accordingly.

[revised manuscript text omitted]

---

## Author Response (AR2)

*In this document, the reviewer comments are in black, the authors responses are in red.*

The authors thank the reviewer for their additional comments.

SUMMARY:
This paper discusses the analyses of wind turbine wake structure considering varying atmospheric stability regimes using measurements from a single scanning lidar; representing an extension of the work of Aiken and Lundquist (JTECH, 2014). Frequency in wake detection, wake velocity deficits, wake width, and wake centerline results are presented. I believe the effort the authors have made through this revision iteration to elaborate on many of the experiment details and analysis methodologies has greatly improved this manuscript, and the authors effectively addressed the suggested revision edits. I accept the current version for publication, though have just a couple remaining technical revision requests.

Thank you for valuing our response to the review!

MINOR COMMENTS/QUESTIONS:
1.  P16, L5-6: The text references Figure 12-a and 12-b, but the actual Figure 12 (and its caption) on P17 do not reference a or b, rather left and right.

Thank you for noticing this incongruity. We have added labels a) and b) to Figure 12:

[Figure]

The caption of the Figure has been modified accordingly: "Qualitative cross-stream slices of stream-wise velocity deficit. The perspective is looking downwind. a) a graphic representation of the classical Gaussian shape of the magnitude of wake velocity deficit. b) an ellipsoid which represents the vertical stretching of the 3-D structure of a turbine wake as a consequence of wind veer."

2.  P 16, L10: The statement "…the clockwise change of wind direction with height that occurs in strongly stable conditions…" implies that such a change in wind direction

always occurs in strongly stable conditions, and perhaps implies that it does not occur in unstable conditions. First, depending on the depth being considered, there is commonly wind veer through some depth of the boundary layer, stability aside. Second, the magnitude of wind veer through the relevant heights of a typical utility scale turbine rotor can vary considerably in strongly stable conditions, and isn't necessarily always present. I would encourage the authors to reconsider the wording used here.

We agree with the reviewer about the general complexity of wind veer. We have rephrased the sentence as: "…the clockwise change of wind direction with height that _often_ occurs in stable conditions…" to emphasize that the wind veer is systematically more common during stable conditions, which are the one we analyze in our study.

[revised manuscript text omitted]